# DualTime: A Dual-Adapter Language Model for Time Series Multimodal Representation Learning

## Abstract

The recent rapid advancements in language models (LMs) have garnered attention in time series multimodal representation learning. However, existing contrastive learning-based and prompt-based LM approaches tend to be biased, often assigning a primary role to time series modality while treating text modality as secondary. We classify these approaches under a temporal-primary paradigm, which overlooks the unique and critical task-relevant information provided by the text modality, failing to fully leverage mutual benefits and complementarity of different modalities. To fill this gap, we propose a novel textual-temporal multimodal learning paradigm that enables either modality to serve as the primary one while being enhanced by the other, thereby effectively capturing modality-specific information and fostering cross-modal interaction. In specific, we design *DualTime*, a language model composed of dual adapters to implement temporal-primary and textual-primary modeling simultaneously. Within each adapter, lightweight adaptation tokens are injected into the top layers of LM to encourage high-level cross-modal interaction. The shared LM pipeline by dual adapters not only achieves adapter alignment but also reduces computation resources and enables efficient fine-tuning. Empirically, DualTime demonstrates superior performance, achieving notable improvements of 7% accuracy and 15% F1 in supervised settings. Furthermore, the few-shot label transfer experiments validate DualTime's expressiveness and transferability.

## 1 Introduction

Time series is a ubiquitous data modality across a wide range of real-world applications Trirat et al. (2024). In recent years, the availability of various modalities (e.g., text Li et al. (2020), images Lalam et al. (2023), sensor data Zurita et al. (2017), graph Liu et al. (2024a)) coupled with traditional time series is increasing. Each modality contains both shared information that overlaps with other modalities and unique information that may provide distinct insights Liang et al. (2024). Jointly modeling time series with other modalities offers richer insights for decision-making. For example, in medical applications, electroencephalogram (EEG) signals capture physiological activity, while clinical records provide health history. Analyzing only symptoms may suggest epilepsy but can't specify seizure types, while EEGs detect abnormal activity but lack personal context. Integrating both modalities can improve diagnostic precision and rationality. A key challenge in time series multimodal learning is to effectively represent and exploit the complementarity and interactions of different modalities Guo et al. (2019).

Recently, large-scale pre-trained language models (LMs) have shown exceptional proficiency in understanding sequential data Chang et al. (2023); Gruver et al. (2024), sparking interest in integrating them into time series multimodal learning Deldari et al. (2022); Ye et al. (2024). Several contrastive learning-based works leverage language models as encoders to extract meaningful representations of text modality, which in turn guide the pre-training of time series encoder but are not present during the inference stage Liu et al.; Yu et al. (2024); King et al. (2023). For instance, METS Li et al. (2024) utilizes a frozen clinical LM to derive embeddings from clinical reports, aligning them with ECG embedding through contrastive learning to enhance ECG signal. And only ECG encoder provides decision for inference. Other prompt-based works not only utilize a frozen LM as a text modality encoder, but also fine-tune another LM as a brain to process the fused multimodal input Jia et al.

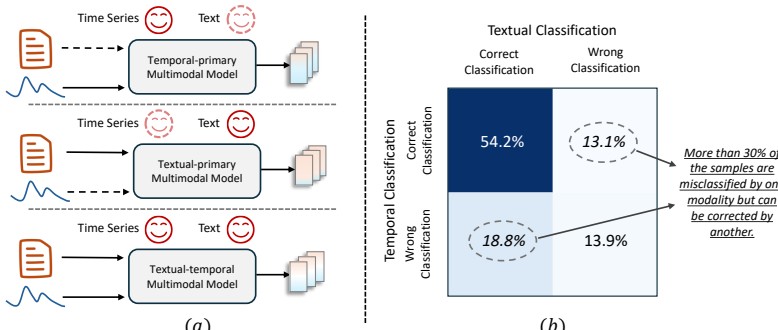

Figure 1: (a) Different time series multimodal modeling paradigms. (b) Unimodal classification results on the PTB-XL dataset (5 classes), using LSTM for temporal classification and BERT for textual classification. The circled samples are misclassified by one modality but corrected by another, demonstrating the complementary information of different modalities.

(2024); Liu et al. (2024b); Cheng et al. (2024); Chan et al. (2024). Specifically, text modality is treated as a prompt of the time series modality to guide LLM's reasoning on the temporal input. For instance, Time-LLM Jin et al. (2023) assembles dataset descriptions, task instructions, and data statistics into a text prompt to facilitate LM's understanding of time series data.

In these LM-based multimodal works, time series is typically considered the primary modality, being more relevant for decision-making, while text serves as an auxiliary modality to enhance the time series embedding, either by projecting textual knowledge into the time series encoder using contrastive learning or by guiding LM with a textual prompt to generate more contextually appropriate responses for temporal inputs. We classify these approaches as temporal-primary multimodal models. However, in some cases, the textual information is no less important than temporal information. As shown in Figure 1 (b), we conduct a unimodal classification experiment on the PTB-XL ECG dataset and find that 18.8% of samples are correctly classified by the text modality but misclassified by the time series modality, while 13.1% shows the reverse. This highlights the complementarity of the two modalities and suggests that the text modality contains even more unique task relevant information. In these cases, viewing text modality as auxiliary may introduce bias and fail to capture essential textual information while a text-primary perspective could enable a more comprehensive understanding of the informative content provided by the text.

To fully exploit the complementarity and mutual benefits of different modalities, we propose a novel textual-temporal multimodal learning paradigm to integrate both temporal-primary and textual-primary perspectives (as shown in Figure 1 (a)). However, to effectively construct LM-based approach of such paradigm is technically non-trivial. The most straightforward solution is to train a LM-based submodel separately for each perspective. Nevertheless, there remain two-fold challenges: First, considering LMs involved, two separately trained submodels suffer non-negligible computational costs. Second, the integration of submodels and the design of single submodels should fuse the two modalities from different perspectives to sufficiently capture both shared and unique information from each modality. Note that the naive multimodal concatenation at LM input layer of existing works is difficult to extract high-level multimodal semantics.

To address the aforementioned challenges, we propose DualTime, a multimodal language model for time series representation learning, consisting of a temporal-primary multimodal adapter and a textual-primary multimodal adapter to effectively explore the complementary information in multimodal input. Under dual adapter design, each modality has the chance to serve as the primary modality and get improved by the other modality. Within each adapter, multimodal fusion is achieved by injecting learnable adaptation tokens into the top layers to extract high-level multimodal semantics. Furthermore, both adapters share the same LM backbone to reduce computational resources. Meanwhile, we keep the majority of LM's parameters frozen to make different modalities benefit from its pre-trained knowledge. We update only a small portion of LM's parameters, adapting it to our task while enabling efficient fine-tuning. In addition, by pipeline sharing, the modality alignment of different adapters could be accomplished. Our main contributions are as below:

- We are the first to propose a textual-temporal multimodal learning paradigm that treats both modalities equally. This paradigm fully leverages the rich complementary semantics of time series and text modality and captures the intricate interaction across different modalities.

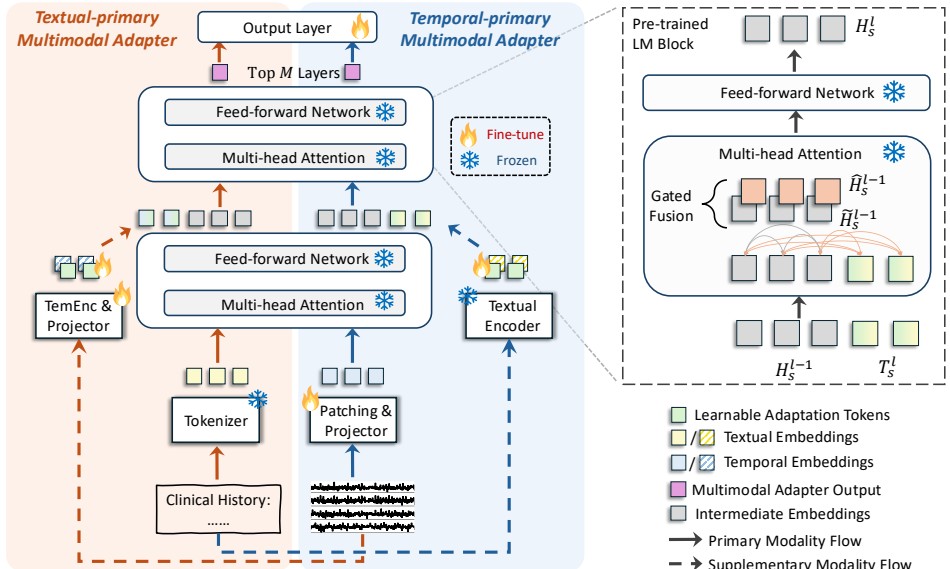

Figure 2: DualTime architecture. It consists of dual adapters to model time series and text as primary modality respectively. Dual adapters share the same LM parameters to reduce computational cost and realize adapter alignment. The LM's pre-trained knowledge is preserved by adopting a zero-initialized gating strategy. The high-level cross-modal fusion is achieved by injecting trainable adaptation tokens in the top layers of LM within each adapter.

- We propose **DualTime**, a dual-adapter language model for time series multimodal representation learning. Each adapter performs the mutual integration of time series and text modalities by introducing learnable tokens into the top layers of the LM backbone, facilitating high-level multimodal semantic fusion. The shared LM pipeline allows both adapters to leverage the pre-trained knowledge and achieves more efficient fine-tuning.

- **DualTime** demonstrates superior performance on public real-world datasets, showing its strong generalization and transferability. Notably, it achieves an average improvement of **7%** in accuracy and **15%** in F1 score under supervised learning. **The code of DualTime is provided in the supplementary materials.**

## 2 METHODOLOGY

In this work, we focus on sample-level time series multimodal data. Specifically, each sample is a time-text pair (e.g., ECG signal and its coupled clinical report). The whole dataset is denoted as $\mathcal{S} = \{(\boldsymbol{X_1}, \boldsymbol{S_1}), (\boldsymbol{X_2}, \boldsymbol{S_2}), ..., (\boldsymbol{X_N}, \boldsymbol{S_N})\}$, where $\boldsymbol{X_i} \in \mathbb{R}^{T \times d}$ denotes a $d$-dimension multivariate time series modality with length $T$ and $\boldsymbol{S_i}$ denotes the paired textual modality. For simplicity, we omit the sample indicator subscript in the following.

In summary, to fully utilize the complementary information of different modalities, DualTime consists of two multimodal adapters, namely a textual-primary multimodal adapter, and a temporal-primary multimodal adapter. Each adapter treats one modality as the primary modality and enhances it with the other modality. Both adapters share the same frozen pre-trained language model with $L$ layers. Each adapter implements multimodal fusion in the topmost $M$ $(M \leq L)$ transformer blocks of the language model. The shared language model backbone facilitates efficient fine-tuning and encourages the dual adapters' embedding space alignment.

### 2.1 TEXTUAL-PRIMARY MULTIMODAL ADAPTER

Processed by the textual tokenizer, the text input can be modeled by $I^s$-length word tokens with embedding $\boldsymbol{E}_s \in \mathbb{R}^{I^s \times D}$, where $D$ is the hidden dimension. For the first $L - M$ transformer layers, they are standard transformer layers. The forward process of layer-$l$ is:

$$\tilde{\boldsymbol{H}}_s^{l-1} = \text{LN}\left(\text{MHA}\left(\boldsymbol{W}_q^l \boldsymbol{H}_s^{l-1}, \boldsymbol{W}_k^l \boldsymbol{H}_s^{l-1}, \boldsymbol{W}_v^l \boldsymbol{H}_s^{l-1}\right)\right) + \boldsymbol{H}_s^{l-1}, \quad (1)$$

$$\boldsymbol{H}_s^l = \text{LN}\left(\text{MLP}\left(\tilde{\boldsymbol{H}}_s^{l-1}\right)\right) + \tilde{\boldsymbol{H}}_s^{l-1}, \quad (2)$$

where $\boldsymbol{H}_s^l$ is the output of layer-$l$ with $\boldsymbol{H}_s^0 = \boldsymbol{E}_s$, MHA, LN, MLP denote the multi-head attention, the layer normalization, and the multi-layer perception, respectively. To obtain the query, key, value matrics at layer-$l$, $\boldsymbol{W}_q^l, \boldsymbol{W}_k^l, \boldsymbol{W}_v^l$ are parameterized by the pre-trained language model. Meanwhile, the attention operation $\mathrm{Attention}$ is defined by:

$$\mathrm{Attention}\left(\boldsymbol{Q}, \boldsymbol{K}, \boldsymbol{V}\right) = \mathrm{softmax}\left(\boldsymbol{Q}\boldsymbol{K}^T / \sqrt{d_k}\right)\boldsymbol{V}, \tag{3}$$

where $\boldsymbol{Q}, \boldsymbol{K}, \boldsymbol{V}$ are corresponding query, key, and value matrices, $d_k$ is the dimension of key.

Furthermore, we follow the adapter architecture in Zhang et al. (2023a) and utilize a lightweight adapter mechanism to achieve multimodal modeling at the topmost $M$ transformer blocks. Specifically, we adopt learnable length-$P$ adaptation tokens $\boldsymbol{T}_s^l$ at each multimodal fusion layer $l \left(L - M + 1 \le l \le L\right)$, where the adaptation tokens $\boldsymbol{T}_s^l \in \mathbb{R}^{P \times D}$ have the same dimension as language model. As to the secondary temporal modality, a trainable temporal encoder and a cross-modal projector are utilized to transform the time series input into the language model embedding space:

$$\boldsymbol{Z}_s = \mathrm{Projector}\left(\mathrm{TemEncoder}\left(\boldsymbol{X}\right)\right). \tag{4}$$

The temporal encoder can be any time-series encoder that best fits the specific datasets, while the projector is a linear layer responsible for dimension transformation. For decreasing the computational cost, different multimodal fusion layers will share the same temporal embedding. Thus, the adaptation tokens of textual-primary multimodal adapter will be calculated by:

$$\tilde{\boldsymbol{T}}_s^l = \boldsymbol{T}_s^l + \boldsymbol{Z}_s. \tag{5}$$

For the topmost $M$ transformer layers, the multimodal forward process is formalized as:

$$\tilde{\boldsymbol{H}}_s^{l-1} = \mathrm{LN}\left(\mathrm{MHA}\left(\boldsymbol{W}_q^l \boldsymbol{H}_s^{l-1}, \boldsymbol{W}_k^l \boldsymbol{H}_s^{l-1}, \boldsymbol{W}_v^l \boldsymbol{H}_s^{l-1}\right)\right) + \boldsymbol{H}_s^{l-1}, \tag{6}$$

$$\hat{\boldsymbol{H}}_s^{l-1} = \mathrm{LN}\left(\mathrm{MHA}\left(\boldsymbol{W}_q^l \boldsymbol{H}_s^{l-1}, \boldsymbol{W}_k^l \tilde{\boldsymbol{T}}_s^l, \boldsymbol{W}_v^l \tilde{\boldsymbol{T}}_s^l\right)\right) + \boldsymbol{H}_s^{l-1}, \tag{7}$$

$$\boldsymbol{H}_s^l = \mathrm{LN}\left(\mathrm{MLP}\left(Gate^l \hat{\boldsymbol{H}}_s^{l-1} + \tilde{\boldsymbol{H}}_s^{l-1}\right)\right) + \left(Gate^l \hat{\boldsymbol{H}}_s^{l-1} + \tilde{\boldsymbol{H}}_s^{l-1}\right). \tag{8}$$

In particular, combined with the pre-trained projection matrices $\boldsymbol{W}_k^l, \boldsymbol{W}_v^l$, the learnable adaptation tokens will serve as key, value matrices of the multi-head attention layer. In Equation (8), we perform a zero-initialized gating strategy to achieve multimodal adaptation token fusion Zhang et al. (2023a). Gating parameter $Gate^l$ will be initialized as zero at the beginning of training, the multimodal adaptation tokens will be injected gradually, which can preserve the pre-trained knowledge and capacities of LMs.

## 2.2 TEMPORAL-PRIMARY MULTIMODAL ADAPTER

Considering the sequential property of time series, the temporal-primary multimodal adapter takes the time series data as the language model input. We utilize the common patching strategy for time series modeling in related works Nie et al. (2022); Zhou et al. (2024). Several adjacent timestamps will be assembled as a token, which can provide local semantic information within a time series. For a pre-defined patch size $p$ and stride $s$, the time series input $\boldsymbol{X} \in \mathbb{R}^{T \times d}$ can be reorganized as $\tilde{\boldsymbol{X}} \in \mathbb{R}^{T_s \times (p \times d)}$, where $T_s = \left\lceil \frac{T-p}{s} \right\rceil + 1$ is the number of temporal tokens. Subsequently, we utilize a projector (i.e. linear layer) to adjust the dimension of temporal tokens. The adjusted temporal token can be denoted as $\boldsymbol{E}_t \left(\boldsymbol{E}_t \in \mathbb{R}^{T_s \times D}\right)$.

With $\boldsymbol{H}_t^0 = \boldsymbol{E}_t$ as the input of the first transformer layer, the model forward process will be similar to the ones introduced in Section 2.1, e.g., Equation (1-2) and Equation (5 - 8).

Differently, for the secondary text input, we use a pre-trained BERT Devlin et al. (2018) model as a text encoder (similar to the temporal encoder in Equation (4)) to extract textual information:

$$\boldsymbol{Z}_t = \mathrm{Proj}\left(\mathrm{BERT}\left(\boldsymbol{S}\right)\right). \tag{9}$$

## 2.3 PRE-TRAINED LANGUAGE MODEL PARAMETERS SHARING

Aided by our dual adapter model design, most of the pre-trained language model parameters (e.g., the attention weight matrices $\boldsymbol{W}_q, \boldsymbol{W}_k, \boldsymbol{W}_v$, and the MLP layer of each transformer block) could be shared by both textual-primary multimodal adapter and temporal-primary multimodal adapter. On the one hand, the frozen parameters could preserve the knowledge and sequential modeling capacities of the language model. On the other hand, since most of the parameters in our proposed adapters are shared, there is only a minimal increase in the training parameters compared to a single adapter. This ensures complementary modeling between the two modalities while still allowing for efficient fine-tuning. Additionally, by sharing the same LM pipeline, the embedding spaces of different adapters are easily aligned, further facilitating the integration of dual adapters.

## 2.4 TRAINING LOSS

**Supervised Learning.** For supervised classification, we add the last transformer layer output of each adapter together to obtain the final multimodal representation. Then, an extra linear classifier and the cross-entropy loss are used for supervised training.

**Unsupervised Representation Learning.** For unsupervised representation learning, we adopt the contrastive learning paradigm. In particular, for data augmentation, we add random Gaussian noise to the original input. The noise-corrupted sample and its original sample are a positive pair within each adapter. We denotes $\boldsymbol{H}_s'^L$ as the augmentation of $\boldsymbol{H}_s^L$, and $\boldsymbol{H}_t'^L$ as the augmentation of $\boldsymbol{H}_t^L$. The contrastive loss could be divided into two parts, within-adapter contrastive loss and cross-adapter contrastive loss.

Formally, by maximizing the agreement between positive pairs and minimizing the similarity between negative pairs (i.e., different input instances), in a mini-batch with size $B$, the within-adapter contrastive losses are

$$\mathcal{L}_s = -\sum_{i=1}^{B} \log \frac{\exp\left(\text{sim}\left(\boldsymbol{H}_{s,i}^L, \boldsymbol{H}_{s,i}'^L\right)/\tau\right)}{\sum_{k=1}^{B} \mathbb{1}_{[k \neq i]} \exp\left(\text{sim}\left(\boldsymbol{H}_{s,i}^L, \boldsymbol{H}_{s,k}^L\right)/\tau\right)}, \quad \mathcal{L}_t = -\sum_{i=1}^{B} \log \frac{\exp\left(\text{sim}\left(\boldsymbol{H}_{t,i}^L, \boldsymbol{H}_{t,i}'^L\right)/\tau\right)}{\sum_{k=1}^{B} \mathbb{1}_{[k \neq i]} \exp\left(\text{sim}\left(\boldsymbol{H}_{t,i}^L, \boldsymbol{H}_{t,k}^L\right)/\tau\right)},$$
(10)

where $\mathbb{1}_{[k \neq i]}$ is the indicator function and $\tau$ is the temperature parameter, $\text{sim}(\cdot, \cdot)$ is the dot product between two $\ell_2$-normalized vectors.

The cross-adapter contrastive learning assumes that the embeddings from two adapters for one temporal-textual input pair should be similar. Concurrently, embedding from different instances should be considered negative pairs. In this vein, the cross-adapter contrastive loss is given by:

$$\mathcal{L}_{cross} = -\sum_{i=1}^{B} \left( \log \frac{\exp\left(\text{sim}\left(\boldsymbol{H}_{s,i}^L, \boldsymbol{H}_{t,i}^L\right)/\tau\right)}{\sum_{k=1}^{B} \mathbb{1}_{[k \neq i]} \exp\left(\text{sim}\left(\boldsymbol{H}_{s,i}^L, \boldsymbol{H}_{t,k}^L\right)/\tau\right)} + \log \frac{\exp\left(\text{sim}\left(\boldsymbol{H}_{t,i}^L, \boldsymbol{H}_{s,i}^L\right)/\tau\right)}{\sum_{k=1}^{B} \mathbb{1}_{[k \neq i]} \exp\left(\text{sim}\left(\boldsymbol{H}_{t,i}^L, \boldsymbol{H}_{s,k}^L\right)/\tau\right)} \right).$$
(11)

The overall loss function of unsupervised representation learning is given by:

$$\mathcal{L}_{unsup} = \mathcal{L}_s + \mathcal{L}_t + \mathcal{L}_{cross}.$$
(12)

Note that for the variants of DualTime, namely DualTime (Time) and DualTime (Text), we only adopt the within-adapter contrastive loss for training.

## 3 EXPERIMENTS

The main research questions of this work are **Q1:** How well does DualTime perform in learning high-quality representations with supervision signals? (Section 3.2) **Q2:** How capable is DualTime in generating general representations under unsupervised learning? (Section 3.4) **Q3:** How adaptable is DualTime while conducting few-shot learning? (Section 3.3) Additionally, we conduct experiments on ablation study, textual encoder testing, sensitivity analysis, and efficiency evaluation, providing deeper insights into the model's mechanisms, robustness and superiority.

## 3.1 EXPERIMENTAL SETUP

**Datasets** All experiments are conducted on publicly available real-world multimodal time series datasets: the PTB-XL electrocardiogram (ECG) dataset Wagner et al. (2020) and the TUSZ electroencephalogram (EEG) dataset Shah et al. (2018). (1) PTB-XL [1]: This dataset consists of 12-lead

---

[1] https://physionet.org/content/ptb-xl/1.0.3/

Table 1: **Supervised Learning**. DualTime achieves an average improvement of **7%** in Acc. and **15%** in F1 across all experiments. The best results are in **bold** while the second and third best are in underlined. "Acc.", "Pre.", and "Rec." represent accuracy, precision and recall respectively. All LM-based models are highlighted in grey.

| Modality | Model | PTB-XL Detection | | | | PTB-XL Classification | | | | TUSZ Detection | | | | TUSZ Classification | | | | Average | |
|---|---|---|---|---|---|---|---|---|---|---|---|---|---|---|---|---|---|---|---|
| | | Acc. | Pre. | Rec. | F1 | Acc. | Pre. | Rec. | F1 | Acc. | Pre. | Rec. | F1 | Acc. | Pre. | Rec. | F1 | Acc. | F1 |
| LM-free Model — Time | LSTM | 0.68 | 0.60 | 0.48 | 0.48 | 0.67 | 0.63 | 0.50 | 0.52 | 0.76 | 0.53 | 0.54 | 0.54 | 0.58 | 0.44 | 0.27 | 0.26 | 0.67 | 0.45 |
| | TimesNet | 0.68 | 0.46 | 0.46 | 0.45 | 0.67 | 0.59 | 0.48 | 0.50 | 0.74 | 0.59 | 0.63 | 0.59 | 0.76 | 0.75 | 0.72 | 0.71 | 0.71 | 0.56 |
| | LightTS | 0.68 | 0.59 | 0.53 | 0.54 | 0.59 | 0.46 | 0.44 | 0.45 | 0.74 | 0.53 | 0.53 | 0.54 | 0.71 | 0.72 | 0.58 | 0.58 | 0.68 | 0.53 |
| | Dlinear | 0.68 | 0.58 | 0.50 | 0.49 | 0.61 | 0.46 | 0.41 | 0.41 | 0.78 | 0.52 | 0.52 | 0.52 | 0.71 | 0.62 | 0.60 | 0.59 | 0.70 | 0.50 |
| | Pyraformer | 0.76 | 0.66 | 0.59 | 0.58 | 0.66 | 0.56 | 0.49 | 0.51 | **0.84** | 0.47 | 0.50 | 0.47 | 0.75 | **0.77** | 0.67 | 0.72 | 0.75 | 0.57 |
| | ETSformer | 0.72 | 0.63 | 0.57 | 0.55 | 0.54 | 0.45 | 0.38 | 0.40 | 0.79 | 0.53 | 0.53 | 0.53 | 0.73 | 0.70 | 0.66 | 0.66 | 0.70 | 0.54 |
| | Autoformer | 0.72 | 0.56 | 0.56 | 0.54 | 0.62 | 0.47 | 0.44 | 0.44 | 0.79 | 0.52 | 0.51 | 0.51 | 0.70 | 0.64 | 0.64 | 0.61 | 0.71 | 0.53 |
| | Crossformer | 0.66 | 0.58 | 0.51 | 0.53 | 0.65 | 0.55 | 0.48 | 0.50 | 0.79 | 0.50 | 0.51 | 0.50 | 0.72 | 0.71 | 0.58 | 0.58 | 0.71 | 0.53 |
| | FEDformer | 0.67 | 0.57 | 0.50 | 0.51 | 0.65 | 0.53 | 0.47 | 0.49 | 0.76 | 0.57 | 0.58 | 0.57 | 0.68 | 0.48 | 0.54 | 0.48 | 0.69 | 0.51 |
| | Informer | 0.67 | 0.59 | 0.51 | 0.52 | 0.67 | 0.59 | 0.51 | 0.52 | 0.82 | 0.57 | 0.55 | 0.55 | 0.77 | 0.74 | 0.69 | 0.71 | 0.73 | 0.58 |
| | Reformer | 0.69 | 0.56 | 0.53 | 0.54 | 0.65 | 0.53 | 0.48 | 0.49 | **0.84** | 0.52 | 0.50 | 0.48 | 0.74 | 0.75 | 0.61 | 0.66 | 0.73 | 0.54 |
| | iTransformer | 0.56 | 0.42 | 0.36 | 0.37 | 0.54 | 0.39 | 0.31 | 0.29 | 0.80 | 0.50 | 0.50 | 0.49 | 0.73 | 0.75 | 0.59 | 0.61 | 0.66 | 0.44 |
| | PatchTST | 0.78 | 0.76 | 0.62 | 0.62 | 0.74 | 0.69 | 0.59 | 0.62 | 0.73 | 0.54 | 0.55 | 0.54 | 0.70 | 0.65 | 0.59 | 0.57 | 0.74 | 0.59 |
| LM-based Model — Time | GPT4TS | 0.71 | 0.58 | 0.52 | 0.53 | 0.59 | 0.46 | 0.45 | 0.45 | 0.78 | 0.48 | 0.48 | 0.48 | 0.71 | 0.73 | 0.60 | 0.64 | 0.70 | 0.53 |
| Text | GPT2 | 0.72 | 0.65 | 0.56 | 0.58 | 0.73 | 0.65 | 0.61 | 0.62 | 0.72 | 0.49 | 0.49 | 0.50 | 0.64 | 0.69 | 0.53 | 0.58 | 0.70 | 0.57 |
| | BERT | 0.70 | 0.64 | 0.51 | 0.53 | 0.73 | 0.65 | 0.59 | 0.62 | 0.72 | 0.49 | 0.49 | 0.49 | 0.59 | 0.45 | 0.39 | 0.40 | 0.69 | 0.51 |
| | Llama 3 | 0.73 | 0.60 | 0.60 | 0.60 | 0.74 | 0.69 | 0.56 | 0.55 | 0.72 | 0.63 | 0.63 | 0.63 | 0.66 | 0.62 | 0.47 | 0.47 | 0.71 | 0.54 |
| | ClinicalBERT | 0.73 | 0.57 | 0.54 | 0.53 | 0.74 | 0.69 | 0.56 | 0.55 | 0.72 | 0.65 | 0.68 | 0.66 | 0.67 | 0.36 | 0.64 | 0.43 | 0.72 | 0.54 |
| Time + Text | TimeLLM | 0.69 | 0.60 | 0.48 | 0.47 | 0.67 | 0.59 | 0.46 | 0.48 | 0.75 | 0.51 | 0.51 | 0.51 | 0.69 | 0.70 | 0.50 | 0.47 | 0.70 | 0.48 |
| | UniTime | 0.67 | 0.33 | 0.42 | 0.37 | 0.64 | 0.54 | 0.43 | 0.44 | 0.79 | 0.54 | 0.53 | 0.53 | 0.77 | **0.78** | 0.71 | 0.71 | 0.72 | 0.51 |
| | GPT4MTS | 0.72 | 0.59 | 0.60 | 0.59 | 0.65 | 0.48 | 0.50 | 0.48 | 0.82 | 0.64 | 0.63 | 0.63 | 0.70 | 0.72 | 0.60 | 0.53 | 0.72 | 0.55 |
| | DualTime (Time) | 0.72 | 0.61 | 0.55 | 0.54 | 0.68 | 0.58 | 0.53 | 0.53 | 0.83 | 0.61 | 0.57 | 0.58 | 0.72 | 0.74 | 0.60 | 0.59 | 0.74 | 0.56 |
| | DualTime (Text) | 0.82 | 0.75 | 0.74 | 0.74 | 0.76 | 0.69 | 0.63 | 0.65 | 0.82 | 0.65 | 0.66 | 0.65 | 0.78 | 0.74 | 0.72 | 0.73 | 0.79 | 0.69 |
| | DualTime | **0.83** | **0.77** | **0.75** | **0.76** | **0.80** | **0.74** | **0.73** | **0.73** | **0.84** | **0.69** | **0.69** | **0.69** | **0.79** | 0.77 | **0.80** | **0.78** | **0.82** | **0.74** |

ECG signals, which capture the electrical activity of the heart, along with clinical reports describing signal characteristics without diagnostic labels. PTB-XL provides two label sets: a coarse-grained label set for disease detection (4 classes) and a fine-grained label set for specific disease classification (5 classes). (2) TUSZ v1.5.2 [2]: The Temple University Seizure Corpus (TUSZ) is a large-scale dataset of EEG signals that record the electrical activity of the brain. It includes 19-channel EEG recordings and the clinical history for each patient session. Similar to PTB-XL, TUSZ offers two label sets: a coarse-grained label set for distinguishing seizure and non-seizure EEG signals, and a fine-grained label set for seizure type classification, comprising 5 classes. More details about the datasets, including the label sets, data splits, and preprocessing steps, are provided in Appendix A.1.

**Baselines** Representative baselines are selected to ensure sufficient experiments. (1) **Unimodal LM-free methods**: MLP-based models (LightTS Zhang et al. (2022), DLinear Zeng et al. (2023)); RNN-based models (LSTM Hochreiter and Schmidhuber (1997)); CNN-based models (TimesNet Wu et al. (2022), TS2Vec Yue et al. (2022), TS-CoT Zhang et al. (2023b)); Transformer-based models (Pyraformer Liu et al. (2021), ETSformer Woo et al. (2022), Autoformer Wu et al. (2021), Crossformer Zhang and Yan (2022), FEDformer Zhou et al. (2022), Informer Zhou et al. (2021), Reformer Kitaev et al. (2020), iTransformer Liu et al. (2023), PatchTST Nie et al. (2022), TS-TCC Eldele et al. (2021)). (2) **Unimodal LM-based methods**: BERT Devlin et al. (2018), GPT-2 Radford et al. (2019), GPT4TS Zhou et al. (2024). (3) **Multimodal LM-based methods**: TimeLLM Jin et al. (2023), UniTime Liu et al. (2024b), GPT4MTS Jia et al. (2024) for supervised learning; METS Li et al. (2024), MERL Liu et al. for unsupervised learning. (4) **DualTime variants**: *DualTime (Time)* for temporal-primary multimodal adapter, *DualTime (Text)* for textual-primary multimodal adapter. Note that for GPT-2 or BERT, we use textual embeddings generated by them and then train a linear classifier from scratch for the downstream task.

**Implementations** DualTime adopts a frozen GPT-2 as the backbone. In the textual-primary multimodal adapter, the tokenizer is from GPT-2. To avoid heavy computational costs, we choose a lightweight CNN-based model as temporal encoder, which consists of three conv-blocks and each with three CNN layers. We train it from scratch to adapt it to our tasks. In the temporal-primary multimodal adapter, a frozen BERT serves as a textual encoder. All hidden dimensions are set to 768 to match the dimension of the backbone (i.e. GPT-2). The value of multimodal fusion layers $M$ is 11 and adaptation token length $P$ is 5. Sensitivity analysis of these parameters is in the Appendix A.6. Time series patching size and stride are all 25. Adam is adopted as the optimizer Kingma (2014). All experiments are implemented by PyTorch Framework with a NVIDIA A6000 (48G) GPU.

---

[2]https://isip.piconepress.com/projects/nedc/html/tuh_eeg/

## 3.2 SUPERVISED LEARNING

We add a linear classifier as the output layer of DualTime to verify its ability to learn high-quality representations with supervision signals. As shown in Table 1, **(1)** Time-only models perform better than text-only models, achieving second best in most experiments. PatchTST significantly outperforms other baselines in PTB-XL. This indicates that time series model can better capture decision-relevant information than the textual models on average. **(2)** Compared with text-only BERT and GPT-2, DualTime (Text) enhances text modality with time series data and demonstrates noticeable improvements, underscoring the importance of integrating time series in the textual-primary model. **(3)** Among multimodal approaches based on LMs, UniTime and GPT4MTS exhibit similar performance, outperforming TimeLLM by a 2% accuracy improvement. This performance gap may be due to the differences in their fine-tuning strategies. While TimeLLM relies on a frozen LLM, UniTime and GPT4MTS employ parameter-efficient fine-tuning techniques. **(4)** DualTime significantly outperforms these LM based multimodal methods by 10% accuracy improvement. This discrepancy likely arises from their temporal-primary paradigm, which overlooks critical information in the text modality. In contrast, DualTime integrates both temporal-primary and textual-primary perspectives, allowing for a more comprehensive understanding of the multimodal interactions among different modalities. **(5)** DualTime (Text) generally outperforms DualTime (Time), likely due to the backbone GPT-2's stronger capability in processing text compared to time series. **(6)** DualTime achieves the best performance, improving accuracy by **7%** and F1 by **15%** on average.

## 3.3 FEW-SHOT LEARNING FOR LABEL TRANSFER

To evaluate the transferability of learned representations under supervised learning setting, we introduce a **Few-shot Label Transfer** framework, which facilitates in-dataset transfer between label sets with different granularity (as illustrated in Figure 3). It is common in real-world applications that coarse-grained labels, such as the presence of a disease, are typically easier and less expensive to acquire, whereas fine-grained labels, like specific disease types, often require more effort and resources to obtain. In this framework, we first pre-train the model on a dataset with coarse-grained yet abundant labels (e.g., disease detection) and then fine-tune it using fine-grained but limited labels (e.g., disease classification). More specifically, after supervised learning on coarse-grained dataset, we freeze the pre-trained model parameters and train an additional classifier using limited fine-grained labeled data for few-shot learning. We conduct {5, 10, 15, 20, 50, 100}-shot experiments on all methods and the 5-shot results of DualTime is in Table 2. We further select several competitive baseline methods and show the performance with different shots in Figure 4(b) and leave other baselines in Appendix A.4.

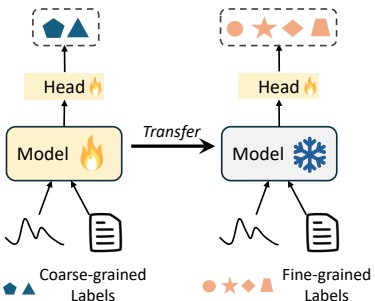

Figure 3: Illustration for **Label Transfer**. We first pre-train a model on dataset with coarse-grained but redundant labels, then fine-tune it on dataset with fine-grained but limited labels.

As shown in Table 2, **(1)** Time-only models generally outperform text-only models. The limited 5-shot time series samples might exhibit patterns captured by time-only models while GPT-2 and BERT struggle to effectively utilize the few available textual samples. **(2)** Additionally, DualTime (Time) surpasses DualTime (Text) on PTB-XL and performs comparably on TUSZ, suggesting that when samples are limited, the time series modality is more important than the text modality. **(3)** Despite training on only 5-shot samples, DualTime outperforms most baselines across nearly all metrics, showcasing its effectiveness in scenarios with limited data. **(4)** As the number of shots ($K$) increases, DualTime's accuracy advantage progressively widens (as depicted in Figure 4(b)).

Table 2: **5-shot Label Transfer**. DualTime achieves almost the best fine-tuning performance, demonstrating its superior few-shot capacity.

| Modality | Model | PTB-XL | | | | TUSZ | | | |
|---|---|---|---|---|---|---|---|---|---|
| | | Acc. | Pre. | Rec. | F1 | Acc. | Pre. | Rec. | F1 |
| Time | LSTM | 0.60 | 0.37 | 0.38 | 0.37 | 0.31 | **0.55** | 0.48 | 0.37 |
| | TimesNet | 0.50 | 0.33 | 0.32 | 0.29 | 0.34 | 0.26 | 0.21 | 0.20 |
| | LightTS | 0.22 | 0.24 | 0.25 | 0.20 | 0.33 | 0.39 | 0.44 | 0.33 |
| | Dlinear | 0.30 | 0.24 | 0.24 | 0.23 | 0.42 | 0.37 | 0.48 | 0.37 |
| | Pyraformer | 0.39 | 0.24 | 0.23 | 0.22 | 0.47 | 0.33 | 0.43 | 0.33 |
| | ETSformer | 0.46 | 0.33 | 0.24 | 0.21 | 0.44 | 0.53 | 0.33 | 0.32 |
| | Autoformer | 0.25 | 0.26 | 0.26 | 0.22 | 0.24 | 0.26 | 0.29 | 0.17 |
| | Crossformer | 0.39 | 0.32 | 0.35 | 0.31 | 0.51 | 0.34 | 0.36 | 0.35 |
| | FEDformer | 0.21 | 0.23 | 0.22 | 0.18 | 0.34 | 0.26 | 0.21 | 0.20 |
| | Informer | 0.47 | 0.35 | 0.35 | 0.34 | 0.24 | 0.33 | 0.21 | 0.17 |
| | Reformer | 0.32 | 0.38 | 0.27 | 0.25 | 0.34 | 0.30 | 0.31 | 0.24 |
| | iTransformer | 0.25 | 0.20 | 0.20 | 0.29 | 0.51 | 0.41 | 0.47 | 0.41 |
| | PatchTST | 0.45 | 0.38 | 0.40 | 0.38 | 0.34 | 0.21 | 0.31 | 0.19 |
| | GPT4TS | 0.20 | 0.20 | 0.20 | 0.18 | 0.45 | 0.42 | 0.49 | 0.38 |
| Text | GPT2 | 0.24 | 0.22 | 0.22 | 0.18 | 0.20 | 0.31 | 0.44 | 0.19 |
| | BERT | 0.45 | 0.34 | 0.33 | 0.32 | 0.24 | 0.35 | 0.32 | 0.24 |
| Time + Text | TimeLLM | 0.49 | 0.28 | 0.33 | 0.30 | 0.29 | 0.33 | 0.26 | 0.25 |
| | UniTime | 0.46 | 0.32 | 0.34 | 0.30 | **0.54** | 0.32 | 0.31 | 0.44 |
| | GPT4MTS | 0.46 | 0.31 | 0.31 | 0.28 | 0.51 | 0.47 | 0.53 | 0.45 |
| | DualTime (Time) | 0.58 | 0.41 | 0.39 | 0.38 | 0.46 | 0.41 | 0.51 | 0.42 |
| | DualTime (Text) | 0.49 | 0.37 | 0.38 | 0.36 | 0.47 | 0.45 | 0.51 | 0.43 |
| | DualTime | **0.64** | **0.52** | **0.50** | **0.50** | 0.52 | 0.48 | **0.56** | **0.48** |

Table 3: **Unsupervised Learning**. 100% labeled data are used for linear classifier training. DualTime achieves an average 2% Acc and 2% F1 improvement, showing its powerful generalization on downstream tasks.

| Modality | Model | PTB-XL Detection Acc. | Pre. | Rec. | F1 | PTB-XL Classification Acc. | Pre. | Rec. | F1 | TUSZ Detection Acc. | Pre. | Rec. | F1 | TUSZ Classification Acc. | Pre. | Rec. | F1 | Average Acc. | F1 |
|---|---|---|---|---|---|---|---|---|---|---|---|---|---|---|---|---|---|---|---|
| **LM-free Model** Time | TSTCC | 0.68 | 0.57 | 0.53 | 0.54 | 0.65 | 0.56 | 0.48 | 0.50 | 0.74 | 0.51 | 0.50 | 0.48 | 0.67 | 0.44 | 0.51 | 0.45 | 0.69 | 0.49 |
| | TS2vec | 0.61 | 0.46 | 0.43 | 0.43 | 0.61 | 0.54 | 0.48 | 0.49 | 0.70 | 0.49 | 0.49 | 0.49 | 0.70 | 0.75 | 0.57 | 0.53 | 0.66 | 0.48 |
| | TSCoT | 0.73 | 0.71 | 0.58 | 0.60 | 0.75 | 0.68 | 0.61 | 0.63 | 0.67 | 0.54 | 0.57 | 0.53 | 0.69 | 0.76 | 0.55 | 0.60 | 0.71 | 0.59 |
| | PatchTST | 0.60 | 0.53 | 0.38 | 0.35 | 0.55 | 0.45 | 0.32 | 0.30 | 0.73 | 0.50 | 0.50 | 0.50 | 0.67 | 0.63 | 0.53 | 0.45 | 0.64 | 0.40 |
| **LM-based Model** Text | GPT2 | 0.72 | 0.65 | 0.56 | 0.58 | 0.73 | 0.65 | 0.61 | 0.62 | 0.72 | 0.49 | 0.49 | 0.50 | 0.64 | 0.69 | 0.53 | 0.58 | 0.70 | 0.57 |
| | BERT | 0.70 | 0.64 | 0.51 | 0.53 | 0.73 | 0.65 | 0.59 | 0.62 | 0.72 | 0.49 | 0.49 | 0.49 | 0.59 | 0.45 | 0.39 | 0.40 | 0.69 | 0.51 |
| Time | METS | 0.74 | 0.66 | 0.57 | 0.58 | 0.71 | 0.64 | 0.57 | 0.60 | 0.65 | 0.55 | 0.59 | 0.53 | 0.57 | 0.46 | 0.26 | 0.20 | 0.67 | 0.48 |
| | MERL | 0.75 | 0.71 | 0.56 | 0.58 | 0.75 | 0.70 | 0.63 | 0.66 | 0.70 | 0.57 | 0.62 | 0.57 | 0.70 | 0.89 | 0.46 | 0.50 | 0.73 | 0.58 |
| Time + Text | DualTime (Time) | 0.68 | 0.52 | 0.46 | 0.44 | 0.60 | 0.48 | 0.39 | 0.40 | 0.68 | 0.52 | 0.52 | 0.51 | 0.66 | 0.50 | 0.66 | 0.49 | 0.66 | 0.46 |
| | DualTime (Text) | 0.72 | 0.66 | 0.55 | 0.57 | 0.73 | 0.66 | 0.63 | 0.64 | 0.70 | 0.50 | 0.50 | 0.50 | 0.70 | 0.58 | 0.77 | 0.60 | 0.71 | 0.58 |
| | DualTime | 0.75 | 0.68 | 0.59 | 0.62 | 0.77 | 0.71 | 0.65 | 0.67 | 0.75 | 0.60 | 0.57 | 0.58 | 0.75 | 0.60 | 0.79 | 0.60 | 0.75 | 0.62 |

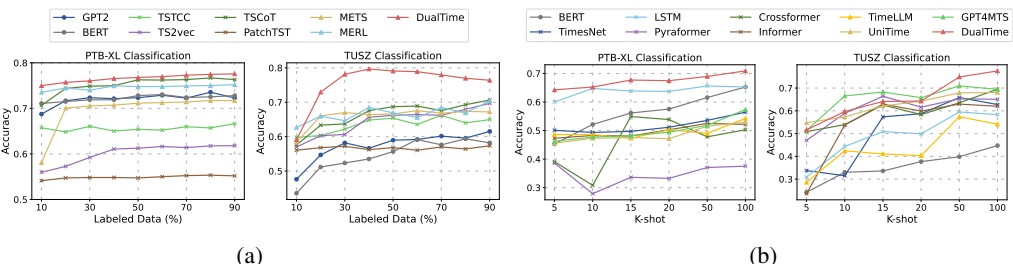

(a)  (b)

Figure 4: (a) Performance comparison for unsupervised representation learning with different proportions of labeled data on classification task. DualTime consistently performs best, especially in TUSZ. (b) Performance comparison for label transfer with different shots. DualTime shows the best performance on nearly all the shots. For small shots, its advantage is not significant while as the shot increases, the performance gap becomes obvious.

## 3.4 UNSUPERVISED LEARNING

To assess our model's ability to generate general representations without ground truth supervision, we conduct unsupervised experiments. Once unsupervised embeddings are obtained for all samples, varying proportions of labeled data, from 10% to 100%, are used to train a linear classifier. Figure 4 (a) illustrates the performance comparison among competitive unsupervised approaches with data proportions ranging from 10% to 90% on two datasets with fine-grained labels. Table 3 shows the results of 100% labeled data proportion. More detailed results can be found in Appendix A.3.

As shown in Table 3, **(1)** Similar to the results of supervised learning, time-only models generally outperform text-only models across all experiments, highlighting the importance of time series data. **(2)** While the multimodal model MERL slightly outperforms the best time-only model TSCoT, METS falls behind, suggesting that multimodal does not always surpass single modality. The effectiveness of multimodal fusion is crucial. **(3)** DualTime surpasses MERL in most experiments, emphasizing the advantages of our complementary textual-temporal multimodal design. **(4)** Overall, DualTime achieves an average accuracy improvement of 2% and consistently outperforms other baselines across varying data proportions in Figure 4 (a). This suggests that the representations learned by DualTime are more expressive and transferable, facilitating effective training even with limited labeled data.

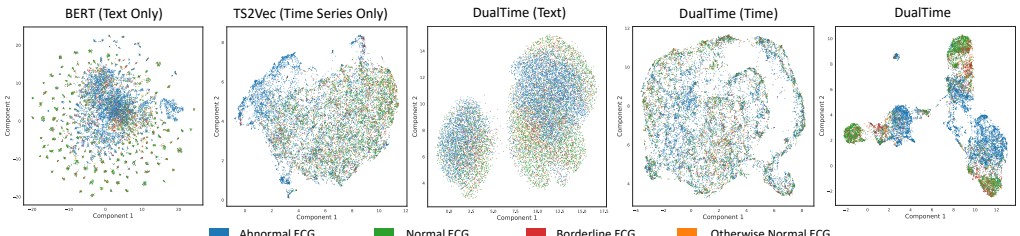

Figure 5: Embedding visualizations of different encoders on PTB-XL, with labels distinguished by color, show that DualTime more clearly separates different classes compared to other models. This demonstrates the effectiveness of our complementary textual-temporal multimodal paradigm.

Table 4: **Influence of Different Textual Encoders**. In general, BERT-based textual encoders demonstrate superior performance, with ClinicalBERT specifically for medical applications achieving the highest average accuracy.

| Textual Encoder | PTB-XL | | | | | | | | TUSZ | | | | | | | | Average | |
|---|---|---|---|---|---|---|---|---|---|---|---|---|---|---|---|---|---|---|
| | Supervised Learning | | | | Unsupervised Learning | | | | Supervised Learning | | | | Unsupervised Learning | | | | | |
| | Classification | | Detection | | Classification | | Detection | | Classification | | Detection | | Classification | | Detection | | | |
| | Acc. | F1 | Acc. | F1 | Acc. | F1 | Acc. | F1 | Acc. | F1 | Acc. | F1 | Acc. | F1 | Acc. | F1 | Acc. | F1 |
| DualTime (BERT) | **0.83** | **0.76** | **0.81** | 0.74 | 0.75 | 0.62 | **0.77** | **0.67** | 0.84 | **0.69** | **0.79** | **0.78** | 0.75 | 0.58 | **0.75** | 0.60 | 0.79 | **0.62** |
| DualTime (RoBERTa) | **0.83** | **0.76** | 0.80 | 0.73 | 0.74 | 0.61 | 0.76 | 0.66 | **0.87** | 0.61 | **0.79** | 0.74 | 0.75 | 0.49 | 0.74 | **0.64** | 0.78 | 0.60 |
| DualTime (ClinicalBERT) | **0.83** | **0.76** | **0.81** | **0.75** | **0.77** | **0.65** | **0.77** | 0.58 | **0.87** | 0.68 | **0.79** | 0.75 | **0.76** | 0.57 | 0.74 | 0.62 | **0.80** | **0.62** |
| DualTime (GPT-2) | 0.82 | 0.75 | 0.80 | 0.73 | 0.74 | 0.60 | 0.76 | 0.66 | 0.86 | 0.52 | 0.72 | 0.60 | 0.71 | 0.56 | 0.68 | 0.49 | 0.76 | 0.58 |

**Visualization** To better visualize the learned representations, we use UMAP McInnes et al. (2018) to project the unsupervised representation learning results into 2D plots. **(1)** Figure 5 displays the embeddings of various encoders on PTB-XL, with labels assigned to different categories. TS2Vec (time-only) successfully identifies abnormal ECGs, while BERT (text-only) performs the worst by mixing all categories, illustrating the advantage of the time series modality. **(2)** Compared with BERT, DualTime (Text) can better distinguish abnormal ECG and normal ECG, indicating the effectiveness of two modalities over one modality. **(3)** Compared with DualTime (Time), DualTime (Text) has obviously better discriminative capacity, supporting the advantage of textual-primary modeling over temporal-primary modeling. **(4)** Overall, DualTime provides the most distinct representations, attributed to the benefit of complementary multimodal modeling.

### 3.5 EXPLORATIONS ON MODEL DESIGN

**Ablation Study** We ablate DualTime into DualTime (Time) and DualTime (Text). Specifically, DualTime (Time) leverages the textual modality to enhance temporal modality modeling, while DualTime (Text) treats the textual modality as primary and the temporal modality as secondary. We evaluate their performances under all three settings, as shown in Table 1, 3, 2. **(1)** Generally speaking, DualTime (Text) has a better performance than DualTime (Time) in supervised learning and unsupervised learning. This suggests that the backbone language model (i.e. GPT-2) demonstrates a better understanding of text compared with time series. **(2)** While DualTime (Time) outperforms DualTime (Text) in PTB-XL 5-shot experiments (as shown in Table 2), possibly because the model lacks sufficient understanding of limited textual data and temporal modality can provide more valuable clues for decision-making. **(3)** Overall, DualTime consistently outperforms single adapter variants, indicating the contributions of both adapters and highlighting the advantages of complementary textual-temporal paradigm over temporal-primary or textual-primary paradigm.

**Multimodal Fusion Gating Analysis** To better understand how multimodal information is integrated within each adapter, we present the multimodal adaptation token fusion gating parameters across different transformer layers in Figure 6. **(1)** At the start of training, there is no multimodal fusion due to the zero-initialized gating strategy. Gradually, the absolute values of the gating parameters gradually increase, indicating a growing level of multimodal fusion. **(2)** We also observe that the gating parameter values are higher in the initial layers (Layer 1 and 2) and the final layers (Layer 10 and 11) compared to the middle layers (Layer 5 and 6). This suggests that the learnable adaptation tokens enhance multimodal integration in initial layers, while deeper layers are likely adapted for different downstream tasks.

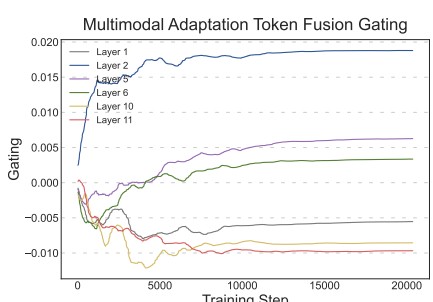

Figure 6: Multimodal gating parameters of different transformer layers.

**Textual Encoder Testing** The current textual encoder used in the temporal-primary adapter of DualTime is BERT. We investigate the impact of various textual encoders by examining the following options: BERT, RoBERTa Liu (2019), ClinicalBERT Wang et al. (2023), and GPT-2. A simplified version of the supervised and unsupervised experimental results are presented in Table 4. More detailed results are in Appendix A.5. As shown in Table 4, BERT-based textual encoders (BERT, RoBERTa, ClinicalBERT) consistently outperform GPT-2. This is likely due to GPT-2's primary focus on text generation, while BERT and its variants excel in comprehending the entire textual input thanks to their masked language model training strategy. Notably, ClinicalBERT specifically pre-trained on medical corpus achieves the highest performance among the tested variants. This underscores the influence of the textual encoder's pre-trained knowledge on its comprehension of

textual modalities. Considering that the textual contents in the PTB-XL and TUSZ datasets are clinical reports, a domain-specific language model tailored for medical applications is more capable of accurately interpreting and analyzing medical textual inputs.

### 3.6 EFFICIENCY EVALUATION

To evaluate the computational costs, we choose the most competitive unimodal baselines (namely TimesNet and PatchTST) and LM-based multimodal approaches (i.e. UniTime and TimeLLM) to compare their efficiency regarding training time per epoch, total parameter size, trainable parameter size, and classification accuracy. Figure 7 shows an efficiency comparison on TUSZ.

Overall, DualTime features a moderate number of trainable parameters while exhibiting the best downstream performance. **(1)** Compared to unimodal methods, DualTime has approximately 1.0 million trainable parameters—larger than PatchTST but significantly smaller than TimesNet, whose complexity arises from its use of 2D convolution operations. **(2)** Additionally, DualTime employs a frozen backbone shared between dual adapters and introduces learnable adaptation tokens, enabling more efficient fine-tuning and effective multimodal fusion. Consequently, DualTime has the smallest parameter count and the shortest training time among multimodal methods, highlighting its efficiency and superior performance.

## 4 RELATED WORK

In this section, we discuss large language models (LLMs) based multimodal works involving both time series and text modalities input. Inspired by Baltrušaitis et al. (2018); Liang et al. (2024), we categorize them into two groups based on how they derive multimodal representation.

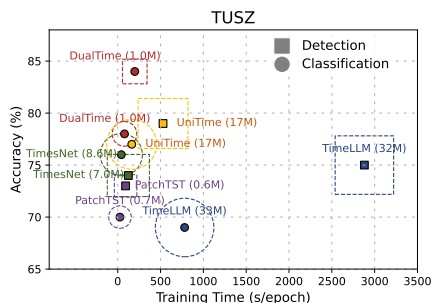

Figure 7: Efficiency comparison on TUSZ. The dotted size represents the model trainable parameter size. DualTime is moderate in size but delivers the performance.

**Coordinated Representation** projects time series and text modality into separate but coordinated spaces, bringing them closer to enforce shared information between modalities Liang et al. (2024). This group, including METS Li et al. (2024), MERL Liu et al., ESI Yu et al. (2024) and King et al. (2023), adopts contrastive learning to align time series and text modalities within a unified space. They leverage LLMs to obtain embedding representations of the text modality, which then guide the pre-training of time series encoder, enhancing the quality and robustness of time series representation. For instance, MERL uses contrastive learning to improve ECG signals under clinical report supervision. However, during training, the contrastive learning often prioritizes shared semantics across modalities, neglecting modality-specific information. In addition, in the inference stage, only the time series modality is present and the text modality is missing. Consequently, such framework depends on time series for decision. The unique and critical task-relevant information from text is overlooked, potentially leading to sub-optimal model performance.

**Joint Representation** projects both modalities into a shared semantic space and fuses them into a single vector Guo et al. (2019). This vector is then fed into into a language model or transformer for prediction. This group includes Time-LLM Jin et al. (2023), UniTime Liu et al. (2024b), GPT4MTS Jia et al. (2024), InstructTime Cheng et al. (2024), MedTsLLM Chan et al. (2024) which implement multimodal fusion by simply concatenating two modalities at the input layer of LLM. However, the order of concatenation influences how LLMs integrate information from different modalities Liu et al. (2024b), resulting in varying cross-modal interactions. Specifically, these works treat the text modality as a prompt prepended to time series modality to facilitate LLM's reasoning on temporal inputs. For instance, UniTime places domain instruction as contextual identifiers before temporal representation to help LLM distinguish between different data sources and adjust its modeling strategy accordingly. However, such sequential concatenation implies that the concatenated modalities are not equally important, making LLM focus more on time series.

All these LLM based multimodal works consider time series as the primary modality for decision-making, with text serving as an auxiliary to enhance time series modeling. In contrast, DualTime allows each modality to act as primary modality through a dual-adapter multimodal language model, which can comprehensively capture the unique and shared semantics provided by different modalities.

## 5 CONCLUSION

In this paper, we propose a new textual-temporal paradigm for time series multimodal learning to delve into the complementary modeling of different modalities. Under this paradigm, we design DualTime with dual adapter design to achieve temporal-primary and textual-primary modeling. Within each adapter, the high-level multimodal fusion is achieved via learnable token injection in the top layers of language model. The pre-trained language model pipeline shared by both adapters enables fine-tuning efficiency. Considering the significant performance gain, the extensive experiments demonstrate that DualTime serves as an effective representation learner in both supervised and unsupervised settings. Regarding the transferability of the model, we demonstrate the superiority of DualTime through few-shot label transfer experiments.

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

# A EXPERIMENTAL DETAILS

## A.1 DATASETS

**Dataset Details** We show the summary of datasets in Table A.1 with dataset statistics and data splitting displayed. For PTB-XL, the coarse-grained labels divide the samples into four classes: *Normal ECG, Borderline ECG, Abnormal ECG, Otherwise normal ECG* Strodthoff et al. (2023), and the fine-grained labels refer to *Normal ECG, Conduction Disturbance, Myocardial Infarction, Hypertrophy*, and *ST/T change*. Similarly, the coarse-grained labels of TUSZ distinguish seizure and non-seizure EEG signals and the fine-grained labels provide further seizure classification: *combined focal (CF) seizures*, *generalized non-specific (GN) seizures*, *absence (AB) seizures*, *combined tonic (CT) seizures*.

Table A.1: Dataset statistics and data split for PTB-XL and TUSZ datasets.

|  | PTB-XL | | TUSZ | |
| --- | --- | --- | --- | --- |
|  | Detection | Classification | Detection | Classification |
| Size of Training Set | 17084 | 17084 | 7766 | 1924 |
| Size of Validation Set | 2146 | 2146 | 5426 | 446 |
| Size of Test Set | 2158 | 2158 | 8848 | 521 |
| Number of Classes | 4 | 5 | 2 | 4 |
| Sequence Length | 1000 | 1000 | 6000 | 6000 |
| Number of Channels | 12 | 12 | 19 | 19 |
| Average Text Length | 13.7 | 13.7 | 24.3 | 23.0 |

**Dataset Examples** PTB-XL dataset contains clinical 12-lead electrocardiograms (ECGs) and their corresponding reports. The clinical reports are automatically generated by the machine and have no diagnosis revealed. TUSZ dataset is the largest EEG seizure database containing 19-channel EEG signals and clinical notes of each patient, for example, clinical history, medications, etc. In this work, we take the clinical history as the experimental textual input. Furthermore, we show two examples for PTB-XL and TUSZ dataset in Figure A.1, respectively. Both time series data and textual data are displayed.

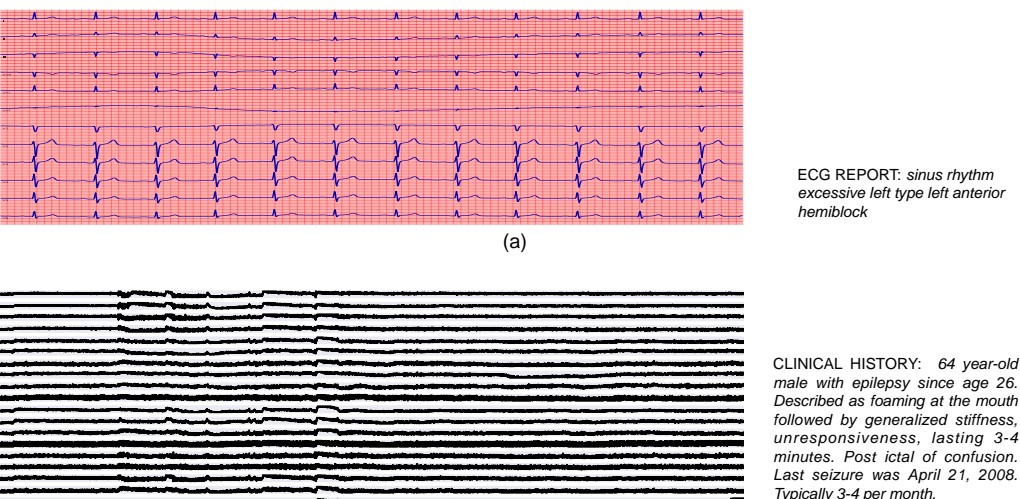

ECG REPORT: *sinus rhythm excessive left type left anterior hemiblock*

(a)

CLINICAL HISTORY: *64 year-old male with epilepsy since age 26. Described as foaming at the mouth followed by generalized stiffness, unresponsiveness, lasting 3-4 minutes. Post ictal of confusion. Last seizure was April 21, 2008. Typically 3-4 per month.*

(b)

Figure A.1: Examples of experimental datasets. (a): PTB-XL dataset collected for electrocardiogram (ECG) analysis. (b): TUSZ dataset collected for electroencephalogram (EEG) analysis.

**Data Pre-processing** All the experiments are conducted on two real-world multimodal time series datasets: PTB-XL Wagner et al. (2020), TUSZ v1.5.2 Shah et al. (2018). PTB-XL contains 12-

lead electrocardiograms (ECGs) with paired clinical reports describing signal characteristics without diagnosis labels. Following Li et al. (2024), all the non-English ECG reports in PTB-XL are translated into English. TUSZ is a large-scale EEG seizure database containing 19-channel EEG signals and clinical history for each session of patients. Following Tang et al. (2021), we process TUSZ to obtain 60-second EEGs for experiments. To avoid data imbalance, we randomly sample at most 8 normal EEGs per patient for training. Both datasets offer two sets of labels: a coarse-grained label set for disease detection and a fine-grained label set for disease classification.

## A.2 EVALUATION METRICS

The evaluation metrics we consider in this paper include accuracy, precision, recall, f1-score. The calculation of these metrics is as follows. For multi-class classification, we report the macro average results.

- **Accuracy**:

$$\text{Accuracy} = \frac{TP + TN}{TP + TN + FP + FN}$$

- **Precision**:

$$\text{Precision} = \frac{TP}{TP + FP}$$

- **Recall**:

$$\text{Recall} = \frac{TP}{TP + FN}$$

- **F1 Score**:

$$F1 = 2 \cdot \frac{\text{Precision} \times \text{Recall}}{\text{Precision} + \text{Recall}}$$

Here, *TP, TN, FP,* and *FN* represent *True Positives, True Negatives, False Positives,* and *False Negatives*, respectively.

## A.3 UNSUPERVISED LEARNING

### A.3.1 UNSUPERVISED BASELINES

For the unsupervised baselines, we follow the codes in original papers to conduct our experiments. Here is a more detailed introduction. TS2Vec is a universal framework based on contrastive learning designed for learning representations of time series at arbitrary semantic levels. TSTCC is an unsupervised time-series representation learning framework that leverages temporal and contextual contrasting to extract meaningful representations from unlabeled data. TSCoT employs co-training based contrastive learning to derive representations through time series prototypes. PatchTST, a Transformer-based model, supports both time series forecasting and self-supervised representation learning and we implement its self-supervised code. METS and MERL utilize contrastive learning to align time series and text modalities without requiring ground truth labels. The aligned time series embeddings are then used to train downstream classifiers. For BERT and GPT-2, we extract textual embeddings generated by these pre-trained language models as general-purpose representations.

### A.3.2 FULL UNSUPERVISED LEARNING RESULTS

The complete unsupervised results of the representative methods, evaluated by training a linear classifier on labeled data subsets ranging from 10% to 90%, are shown in Figure A.2. A simplified version of these results appears in the main text as Figure 4(a). The results cover both classification and detection tasks across two datasets. Notably, DualTime consistently outperforms other methods across varying proportions of labeled data, with its performance remaining stable as the proportion changes. This suggests that the representations learned by DualTime generalize well, allowing effective classifier training even with very few labeled samples.

## A.4 FEW-SHOT LEARNING

The full few-shot results with all the baseline methods compared will be shown in Figure A.3, whose corresponding simplified figure in the main text is Figure 4(b).

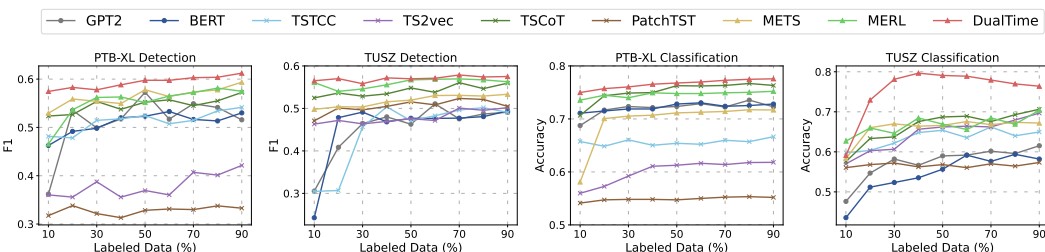

Figure A.2: Performance comparison for unsupervised representation learning with different proportions of labeled data. DualTime consistently performs best, especially in TUSZ classification perhaps due to the beneficial seizures history of patients.

Generally speaking, all models' classification accuracy generally shows a continuous growth trend as the setting of few-shot (K) increases. In particular, under conditions of few-shot scenarios with very limited samples (for example, 5-shot), the transfer performance of text encoders tends to be poor. We might attribute this to the fact that text encoders are trained in large, content-rich text corpora. Although they possess relatively general encoding capabilities, achieving good linear classification results in few-shot scenarios is challenging. The temporal models show different behaviors on different datasets. For PTB-XL dataset, RNN-based models perform well, but former-based methods are more capable for TUSZ's label transfer. On the other hand, our proposed DualTime consistently outperforms the baseline methods on both two datasets. Even with a limited number of available training samples, our model is still able to achieve good classification performance. It substantiates that powered by language model and multimodal input, DualTime demonstrates effectiveness and robust transferability.

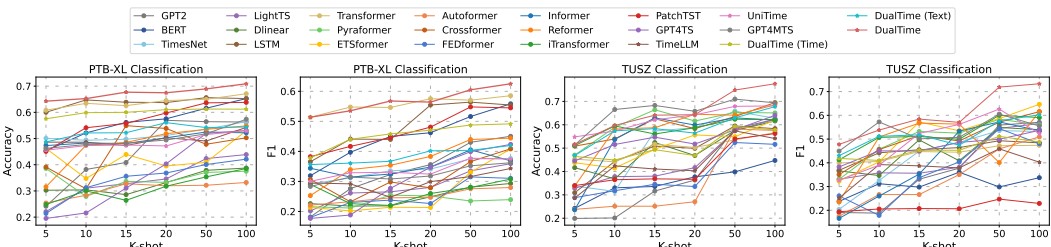

Figure A.3: Full results for label transfer with different few-shot settings.

Table A.2: Supervised learning of disease detection and classification on PTB-XL dataset.

|  | Detection | | | | Classification | | | |
|---|---|---|---|---|---|---|---|---|
|  | Accuracy | Precision | Recall | F1 | Accuracy | Precision | Recall | F1 |
| **DualTime (BERT)** | 0.83 | 0.77 | 0.75 | 0.76 | 0.81 | 0.75 | 0.74 | 0.74 |
| **DualTime (RoBERTa)** | 0.83 | 0.77 | 0.75 | 0.76 | 0.80 | 0.75 | 0.73 | 0.73 |
| **DualTime (ClinicalBERT)** | 0.83 | 0.78 | 0.75 | 0.76 | 0.81 | 0.75 | 0.75 | 0.75 |
| **DualTime (GPT-2)** | 0.82 | 0.76 | 0.74 | 0.75 | 0.80 | 0.74 | 0.73 | 0.73 |

## A.5 TEXTUAL ENCODERS TESTING

We discuss the influence of different textual encoders by considering the following variants: BERT Devlin et al. (2018), RoBERTa Liu (2019), ClinicalBERT Wang et al. (2023), and GPT-2 Radford et al. (2019) as the DualTime textual encoder. The supervised and unsupervised experimental results are reported in the following Table A.2, Table A.3, Table A.4 and A.5. We observe that the BERT-based textual encoders (BERT, RoBERTa, ClinicalBERT) outperform GPT-2. This is likely because GPT-2 is more suited for text generation, while BERT and its variants have a better understanding of the whole textual input due to their masked language model design. Among the variants, ClinicalBERT, which is specifically developed for clinical notes, achieves the best performance.

Table A.3: Unsupervised learning of disease detection and classification on PTB-XL dataset.

| | Detection | | | | Classification | | | |
|---|---|---|---|---|---|---|---|---|
| | Accuracy | Precision | Recall | F1 | Accuracy | Precision | Recall | F1 |
| **DualTime (BERT)** | 0.75 | 0.68 | 0.59 | 0.62 | 0.77 | 0.71 | 0.65 | 0.67 |
| **DualTime (RoBERTa)** | 0.74 | 0.69 | 0.58 | 0.61 | 0.76 | 0.69 | 0.65 | 0.66 |
| **DualTime (ClinicalBERT)** | 0.77 | 0.71 | 0.62 | 0.65 | 0.77 | 0.70 | 0.66 | 0.58 |
| **DualTime (GPT-2)** | 0.74 | 0.67 | 0.58 | 0.60 | 0.76 | 0.68 | 0.64 | 0.66 |

Table A.4: Supervised learning of disease detection and classification on TUSZ dataset.

| | Detection | | | | Classification | | | |
|---|---|---|---|---|---|---|---|---|
| | Accuracy | Precision | Recall | F1 | Accuracy | Precision | Recall | F1 |
| **DualTime (BERT)** | 0.84 | 0.69 | 0.69 | 0.69 | 0.79 | 0.77 | 0.80 | 0.78 |
| **DualTime (RoBERTa)** | 0.87 | 0.76 | 0.59 | 0.61 | 0.79 | 0.77 | 0.74 | 0.74 |
| **DualTime (ClinicalBERT)** | 0.87 | 0.75 | 0.65 | 0.68 | 0.79 | 0.82 | 0.74 | 0.75 |
| **DualTime (GPT-2)** | 0.86 | 0.79 | 0.53 | 0.52 | 0.72 | 0.76 | 0.61 | 0.60 |

## A.6 SENSITIVITY ANALYSIS

As shown in Figure A.4, the performance of our model tends to improve with an increase in the number of multimodal fusion layers. While the length of adaptation tokens has a relatively small impact. Compared to adaptation token length $P$, the influence of multimodal fusion layers $M$ is more evident.

## A.7 FUSION STRATEGY OF DUALTIME

We conduct experiments on different fusion strategies for the auxiliary and primary modalities within each adapter. The table below A.6 presents the experimental results .

It can be observed that dynamic fusion through learnable adaptation tokens achieved the best performance, with an average accuracy of 82%. In contrast, simple concatenation had the poorest performance, with an average accuracy of 75%. likely because it is a static method without learnable parameters, leading to weak generalization capabilities.

The attention mechanism demonstrated the second-lowest performance, achieving an average accuracy of 77%. While it improves upon simple concatenation by introducing a self-attention mechanism, it treats modality tokens almost equally, failing to emphasize the primary and secondary modalities effectively. This lack of distinction causes the textual-primary module and temporal-primary module become similar, making it more challenging for the model to extract the unique information contributed by each modality.

Weighted fusion performed second-best achieving 78% accuracy , perhaps because it can adaptively determine which modality is more important. However, weighted fusion may prioritize one modality over the other, potentially reducing the model's ability to fully extract valuable information from the less prioritized modality. This imbalance could limit the fusion's effectiveness in scenarios where both modalities contribute complementary and unique information. In contrast, the use of learnable adaptation tokens in two modules enforces a distinction between the primary and secondary modalities, guiding the model to focus more effectively on the primary modality. This approach helps the model learn non-overlapping information from each modality, leading to superior performance.

Table A.5: Unsupervised learning of disease detection and classification on TUSZ dataset.

| | Detection | | | | Classification | | | |
|---|---|---|---|---|---|---|---|---|
| | Accuracy | Precision | Recall | F1 | Accuracy | Precision | Recall | F1 |
| **DualTime (BERT)** | 0.75 | 0.60 | 0.57 | 0.58 | 0.75 | 0.60 | 0.79 | 0.60 |
| **DualTime (RoBERTa)** | 0.75 | 0.51 | 0.51 | 0.49 | 0.74 | 0.72 | 0.65 | 0.64 |
| **DualTime (ClinicalBERT)** | 0.76 | 0.58 | 0.57 | 0.57 | 0.74 | 0.70 | 0.65 | 0.62 |
| **DualTime (GPT-2)** | 0.71 | 0.60 | 0.56 | 0.56 | 0.68 | 0.61 | 0.52 | 0.49 |

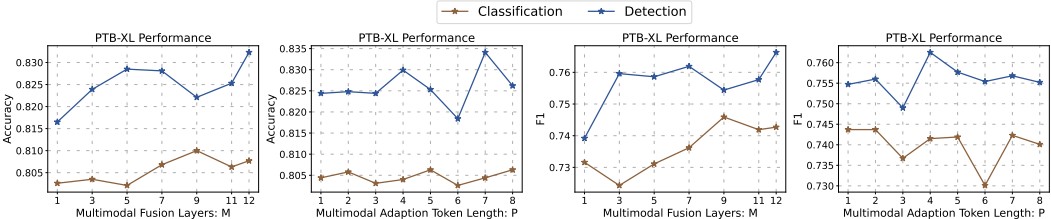

Figure A.4: Hyperparameter study of multimodal fusion layers $M$ and length of adaptation tokens $P$.

Table A.6: Fusion Strategy of Primary Modality and Auxiliary Modality

| DualTime | PTB-XL | | | | | | | | TUSZ | | | | | | | | Average | |
| | | Detection | | | | Classification | | | | Detection | | | | Classification | | | | |
| | Acc. | Pre. | Rec. | F1 | Acc. | Pre. | Rec. | F1 | Acc. | Pre. | Rec. | F1 | Acc. | Pre. | Rec. | F1 | Acc. | F1 |
|---|---|---|---|---|---|---|---|---|---|---|---|---|---|---|---|---|---|---|
| **Adaptation Tokens** | 0.83 | 0.77 | 0.75 | 0.76 | 0.81 | 0.75 | 0.74 | 0.74 | 0.84 | 0.69 | 0.69 | 0.69 | 0.79 | 0.77 | 0.80 | 0.78 | 0.82 | 0.74 |
| **Simple Concatenation** | 0.76 | 0.72 | 0.60 | 0.62 | 0.73 | 0.67 | 0.58 | 0.61 | 0.79 | 0.64 | 0.62 | 0.63 | 0.72 | 0.66 | 0.53 | 0.55 | 0.75 | 0.60 |
| **Attention Mechanism** | 0.77 | 0.72 | 0.61 | 0.63 | 0.75 | 0.70 | 0.64 | 0.66 | 0.79 | 0.66 | 0.65 | 0.65 | 0.75 | 0.72 | 0.63 | 0.59 | 0.77 | 0.63 |
| **Weighted Fusion** | 0.79 | 0.74 | 0.65 | 0.69 | 0.76 | 0.71 | 0.63 | 0.66 | 0.81 | 0.68 | 0.67 | 0.67 | 0.78 | 0.80 | 0.72 | 0.75 | 0.78 | 0.69 |

## B DISCUSSION ABOUT MORE MODALITIES

Here, we discuss the extensibility of the core idea behind DualTime. While DualTime is primarily designed for the time-series and text pair modality, our proposed textual-temporal multimodal learning paradigm, which treats modalities equally, can be extended to other combinations of two modalities or even to scenarios involving more than two modalities.

For instance, some industrial scenarios can collect time series data generated by various sensors as well as images generated by industrial cameras for identifying potential product defects. In these cases, image modality can replace the text modality while time series modality remains unchanged. A similar framework can be designed to combine these two modalities by using a pre-trained vision model, such as ViT, as the encoder for the image modality, and replacing the language model backbone with a large visual pre-trained model.

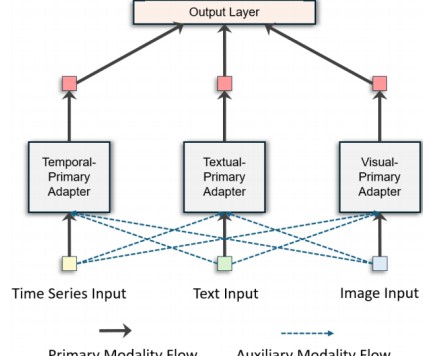

Figure A.5: Illustration of TripleTime

Further, such idea can be extended to more than two modalities. For instance, in addition to time series signals and textual operating logs, images from industrial cameras can help us identify potential defects. This scenario requires for the design of a "TripleTime" model. We can utilizes three adapters to consider multiple modalities simultaneously. As shown in Figure A.5, each adapter will have one primary modality and take the other two modalities as auxiliary inputs. Specifically, GPT-2-based adapters can be used for both temporal and textual inputs, while a pre-trained vision model can serve as the backbone for the visual-primary adapter. Learnable adaptation tokens will inject information from the other two modalities into the primary adapter. Thus, "TripleTime" can achieve simultaneous multimodal modeling for three different modalities.

## C LIMITATIONS AND FUTURE WORKS

One limitation of our work is that, due to the availability of multimodal data, we have only been able to test our model on EEG and ECG datasets within the healthcare domain. For future work, we aim to incorporate additional multimodal datasets from other domains to evaluate the effectiveness and robustness of our model.

Another limitation is that our model can not handle datasets that have varying time series input lengths and channel configurations, which affects its ability to assess transferability across datasets with different settings. Additionally, our use of a data-specific linear output layer for classification limits

the model's capability for zero-shot learning across datasets with different class numbers or label semantics. In future work, we plan to address these issues to improve the cross-dataset transferability of our framework.

## D  SOCIAL IMPACT

Our work focuses on leveraging large language models (LLMs) for multimodal learning in the context of time series analysis. From a narrow perspective, this work can significantly enhance performance with minimal additional cost in domains where time series data are paired with corresponding text, such as patients' diagnostic time series with text reports, machine vibration signals with text logs, or company stock prices with financial reports. From a broader perspective, our approach is adaptable to other modalities and can easily extend to scenarios involving multiple (2+) modalities. Please refer to the Discussion subsection. All in all, our research integrates multiple modalities effectively and efficiently with minimal computation resources, advancing the development of multimodal learning techniques, ultimately contributing to a more intelligent and efficient society.

