# OpenReview forum: "DualTime: A Dual-Adapter  Language Model for Time Series Multimodal Representation Learning"
_ICLR.cc/2025/Conference — Submitted to ICLR 2025_

### Official Review · Reviewer_foTb · 2024-10-30

**Soundness:** 2
**Presentation:** 2
**Contribution:** 2
**Rating:** 5
**Confidence:** 4

**Summary:**

The authors propose DualTime, a method for integrating clinical notes and electrocardiograms into a prediction model. The architecture uses per-modality adapters with time-series-first and text-first representations, which are later merged in a final prediction. The authors also include an unsupervised learning mechanism which should lead to better aligned representations and better predictive performance.

**Strengths:**

- Interesting work on multi-modal time-series that seem to achieve good performance in practice in two datasets.
- Code is shared which means experiments are reproducible.
- Experiments under multiple angles are included: fully supervised, different ratios of unsupervised+supervised data, few-shot learning.

**Weaknesses:**

The paper can be improved across many angles:
- there is notation and formulas that are unnecessary and needed not be there, whereas there are points that could have been clarified in the manuscript (see comments).
- only two datasets are used without an explanation to why that is the case. I find that especially problematic since the paper sells the idea that they do "multi-modal time series learning" whereas the two datasets used include the same modalities.
- baselines have a frozen language model backbone without a clear motivation for why; instead, authors should have included results for the baselines as they were proposed in the literature vs. a version where the LM backbone is frozen (if desired). This choice looks like a severe misrepresentation of baseline models' performance.
- ablation did not cover the case where unsupervised learning is not used at all, and did not include a delta (without unsup. learning vs. with unsup. learning).

**Questions:**

- Interesting way to motivate your work using unimodal classifiers in Figure 1b. Could you please clarify what exactly is the "unimodal classification experiment" (line 79) in one sentence? I.e., what is that you are predicting, and what kind of model / architecture do you use to do that?
- Regarding the unsupervised learning part of your experiments, I missed an ablation of that. You should include experiments where you only use supervised learning, vs. unsupervised plus supervised learning, to show how much you gain from adding these contrastive learning procedure. A delta here ($\Delta$) with and without unsupr. learning would be nice to have.
- You only use two datasets. Why? Does that have to do with the (implicit) assumption you make about the data, that is you have pairs of (clinical notes, ECGs) for each patient throughout time? What about other modalities?
- In lines 92-94, I believe the last two challenges are actually one challenge: that of integrating and fusing diverse multi-modal signals.
- In line 151-152 you state "Each adapter implements multimodal fusion in the topmost M (M ≤ L) transformer blocks of the language model." That is confusing, you don't need to say the M ≤ L part (I had the impresssion for a while that there was a typo). Also, you only adapt the last layer. You could remove all this notation by just saying this, or clearly stating here that $M=1$.
- Are Equations 1, 2, 3 implementing something different from a standard Transformer layer? If it is not, please just state that these layers are standard Transformer layers. If it is, please clearly state somewhere what these differences are (I don't think there are differences).
- For Equations 6, 7, 8, are you using any standard adapter architectures introduced in the literature? If you are, please state that clearly including a citation to the paper. If not, please clearly indicate where your architecture differs from the literature.
- The patch+stride transformation of the time series signal to use in a language model described in lines 197-200 is proposed in this work or was proposed in previous work? If it was proposed in previous work, please state that and add a citation.
- Instead of including equations describing standard Transformer blocks, I believe it would be more relevant to include: an alternative equation 4 where you describe the transformations "TemEnc" and "Proj" in more detail; an equation where you show how $\tilde{X}$ is transformed into $E_t$ (lines 198-200); an alternative equation 9 where you also show what you mean by "Proj" (linear transformation matrix?); there is maybe no need for another separate equation, but add the statement where you show how you compute $\tilde{T}^l_T$ similarly to what you do in Equation 5 for the text-first part of the model.
- Add citations for the datasets in line 258 and explain what you mean by "publicly available". Do you need any credentialing? Is a link to download the datasets available for everyone? Please include the link to download the data, or to the dataset download website.
- Also, please explain what is the difference between an ECG and an EEG and write down what these acronyms mean in the main paper (when describing the datasets). You could also clearly state that are the tasks you have under each dataset in the main paper: at least how many classes do you predict.
- In lines 294-295 you state "Note that we make minimal adjustments to some baselines for our experiments, such as freezing the major parameters of GPT-2 or BERT, only replacing and fine-tuning the output layer for our tasks". Why do you do that? You do not have the baselines you are referring to in the citations, then. This is a major issue, especially if you do not explain why you do this. Do you want to have baseline models with a similar number of trained parameters to your models? Then please add the exact number of trained parameters in each model. I believe you should always include the baselines fully finetuned, if that is how they were proposed in the original work. You may wish to include your frozen baselines as well, but the original baselines should be included. Moreover, if parameter efficiency is an angle in your work, make that clear from the beginning in the introduction (and possibly in the title of the manuscript). You can provide a table comparing the number of trainable parameters and total parameters for each model, including your models and baselines. That would make the picture more complete.
- I think the experiments described in Section 3.3 need much more clarification. You say "First, we train each model in an unsupervised manner." That is too vague. By "which model" do you mean all models evaluated in Fig.3a and in Fig.3b? For the DualTime models that you propose, the way you do that is clear (trained using Eq. 12). For all the other models in Fig.3a and Fig.3b, how did you do that?

---

> ### Author Response · Authors · 2024-11-24
>
> We sincerely appreciate your valuable feedback. Due to space constraints, we cannot provide detailed responses, but we have revised the paper based on your comments and highlighted the changes in red for clarity.
>
> > (0) **Q1&Q4: Questions on Introduction**
>
> **Q1: Unimodal Classifiers**
>
> LSTM is used for temporal classification and BERT for textual classification. The task is to classify five ECG disease types of PTB-XL.
>
>
> **Q4: About the last two challenges**
>
> Both challenges address the integration of multimodal signals but operate at different levels. The first involves fusion of two primary modalities. The second focuses on the fusion of primary modality and secondary modality.
>
> > (1) **W1&Q5&Q6&Q7&Q8&Q9: Notations and Formulas in Methodology**
>
> **Q5: About the Notation M & L**
>
> DualTime's backbone is a pre-trained language model with L transformer blocks. We inject learnable tokens into the topmost M blocks of the backbone. M is a hyperparameter. M=L refers that tokens are injected into all L blocks. M = 1 refers that the tokens are only injected into the last (or topmost) block. Sensitivity analysis of M is in Appendix A.6.
>
> **Q6&Q7: About the Equations 1, 2, 3, 6, 7, 8**
>
> Equations 1, 2, 3 are standard transformer layers.
>
> In equations 6, 7, 8, we utilize adapter architecture introduced in Llama-adapter.
>
> **Q8: Patch+Stride**
>
> The patch+stride is a widely used strategy in time-series modeling. It was first introduced in PatchTST. A citatin has been added.
>
> **Q9: About "TemEnc" and "Proj"**
>
> "TemEnc" refers to the temporal encoder. In theory, users of DualTime can select any time-series encoder that best fits their specific datasets. For our experiments, we opted for a lightweight CNN-based model to balance computational efficiency and performance.
>
> "Proj" in Equation 4 refers to a linear layer used to ensure dimensional alignment. Similarly, the "Proj" mentioned in Equation 9 and in lines 198–200 also represents linear layers responsible for dimension transformations.
>
> > (2) **W2&Q3&Q10&Q11: Datasets & Modalities**
>
> **W2&Q3: Why two datasets**
>
> DualTime which treats modailties equally is particularly suitable for scenarios where temporal and textual modalities are paired one-to-one with closely related information, and each contributes meaningful, decision-relevant information. To date, we have only been able to collect two publicly available datasets that ideally meet these requirements.
>
> **W2&Q3: Data Modality**
>
> In this work, one of our primary motivations is to explore the potential of language models in multimodal learning involving time-series data. While our idea to treat each modality equally—can be extended to other modality pairs, such as time series and images, DualTime cannot be directly applied to a time series–image pair because its backbone is a language model. In future work, we will continue to explore how to more effectively extract complementary information from time-series data and other non-text modalities.
>
> **Q10: Citations and Availability of Datasets**
>
> We have included citations and links to both PTB-XL and TUSZ. PTB-XL is open-access at https://physionet.org/content/ptb-xl/1.0.3/. TUSZ is available at https://isip.piconepress.com/projects/nedc/html/tuh_eeg/, but requires user registration and application for download.
>
> **Q11: Dataset Description**
>
> According to the suggestion, we now modify the 3.1 EXPERIMENTAL SETUP / Dataset section to make it more clear.
>
> > (3) **W3&Q12: Baselines with Frozen Backbone**
>
> In our experiments, we utilize the textual embeddings generated by GPT-2 and BERT and train a linear classifier from scratch for the downstream task. And we do not fully fine-tune GPT-2 or BERT due to the overfitting problem, with the phenomenon being most pronounced on our smallest TUSZ classification dataset. For other LM-based baselines (i.e., GPT4TS, TimeLLM, UniTime, GPT4MTS), we adhered to the fine-tuning strategies outlined in their original papers.
>
> > (4) **W4&Q2&Q13: Unsupervised Learning Experiments**
>
> **Q2:  Unsupervised Experiments**
>
> Our unsupervised training setup follows common practices in time-series unsupervised learning works such as Ts2vec and TSCoT. Contrastive learning is not a contribution of our work. We aim to evaluate our model's unsupervised learning capability compared to the contrastive learning based methods like TS2Vec, TSTCC, TSCoT, METS, and MERL, rather than the advantages of contrastive learning.
>
> **Q13:  Unsupervised Baselines**
>
> Fig. 3a illustrates unsupervised learning, while Fig. 3b focuses on few-shot learning. They are together due to space constraints.
>
> For the contrastive learning baselines, namely TS2Vec, TSTCC, TSCoT, METS, and MERL, we used their original papers' codes for our experiments. For BERT and GPT-2, we extracted textual embeddings from them as general-purpose representations. A detailed introduction to these baselines is provided in Appendix A.3.1.

---

> ### Author Response · Authors · 2024-11-28
>
> To Reviewer foTb:
>
> Thank you very much for your reply! Please let us know if these responses meet your expectations. We are eager to engage in further discussions and continue improving our work.

---

> > ### Comment · Reviewer_foTb · 2024-11-28
> >
> > Dear authors,
> >
> > Thank you for the explanations, and for the rebuttal to my questions. I am increasing my recommendation from 3 reject to 5 marginally below the acceptance threshold.
> >
> > One addditional suggestion I would like to make is:
> > - When you introduce the "unsupervised learning" experiments (Section 3.4), you could clarify a few important things that are now left implicit. First, I suggest to include an Equation in Section 2.4 with the supervised learning loss component. Then, (possibly) in Section 3.1 Experimental setup, you can clearly explain how you train the model in full generality (using task labels).
> > - **Training model with task labels:** That is using a combination of the supervised loss introduced in Section 2.4 plus the unsupervised loss in Equation 12. If you use a scalar to weigh the contribution of each loss component and tune this scalar using held-out validation data, it may be a good idea to include another Equation with this final loss where both supervised+unsupervised components appear, e.g., $\lambda \cdot \mathbb{L}_{sup} + (1 - \lambda) \cdot \mathbb{L}_{unsup}$.
> > - **Training the model without task labels**: you do this only using the unsupervised loss that you already wrote in Equation 12.
> >     - Then, when you explain the "unsupervised learning" experiments' results, I would rephrase that to "representation learning" experiments "by probing the models with varying amounts of labelled data". (Assuming that is the case also for all the baselines)

---

> > > ### Author Response · Authors · 2024-12-01
> > >
> > > Dear Reviewer foTb,
> > >
> > > Thank you very much for your constructive feedback and for updating the score. We have revised our paper according to your comments. While we are unable to update the current PDF, the changes will be reflected in our final manuscript. Below, we outline how we addressed your concerns:
> > >
> > > **S1: the supervised learning loss**
> > >
> > > Following your suggestion, we have added an explicit equation in Section 2.4 to describe the supervised learning loss, specifically the cross-entropy loss.
> > >
> > > $\mathbb{L}*{\mathrm{CE}}=-\sum*{c=1}^C y_c \log \left(\frac{\exp \left(z_c\right)}{\sum_{k=1}^C \exp \left(z_k\right)}\right)$
> > >
> > > where  $\mathbf{z}=\left[z_1, z_2, \ldots, z_C\right]$ is the final represention from the linear classifier. $C$ is the number of labels.
> > >
> > > **S2: Detailed explanation of the training process.**
> > >
> > > Here is a detailed explanation of the training process under two scenarios and we briefly summarize the following explanation in our final manuscript due to limited space.
> > >
> > > - **Training with task labels:** In the experiments described in the supervised learning section, as well as in the linear classifier training of the unsupervised learning section, we utilize cross-entropy loss to guide the model training using task labels.
> > > - **Training without task labels:** In the unsupervised learning section, we train our baselines (TS2Vec, TSTCC, TSCoT, PatchTST, METS, MERL) without ground truth, following their original implementations. For DualTime, we train it using the unsupervised loss described in Section 2.4. After obtaining the unsupervised embeddings from these models, we train a linear classifier from scratch using task labels to evaluate their performance. The unsupervised training and linear head training processes are separate and independent; they do not share the same training objective. This evaluation setting follows prior works such as TS2Vec and TSCoT.
> > >
> > > Here is a more detailed introduction about the unsupervised learning baselines.  TS2Vec is a universal framework based on contrastive learning designed for learning representations of time series at arbitrary semantic levels. TSTCC is an unsupervised time-series representation learning framework that leverages temporal and contextual contrasting to extract meaningful representations from unlabeled data. TSCoT employs co-training based contrastive learning to derive representations through time series prototypes. PatchTST, a Transformer-based model, supports both time series forecasting and self-supervised representation learning and we implement its self-supervised code. METS and MERL utilize contrastive learning to align time series and text modalities without requiring ground truth labels. The aligned time series embeddings are then used to train downstream classifiers.
> > >
> > > **S3: Rephrasing "unsupervised learning experiments" as "representation learning experiments"**
> > >
> > > Based on your suggestion, we have rephrased the "unsupervised learning experiments" section (Section 3.4) as "representation learning experiments." We now explicitly state that these experiments evaluate the learned representations of the baselines by probing the models with varying amounts of labeled data. This adjustment aligns the terminology with the actual goal of the experiments, which is to assess the quality of the representations learned by these baselines. Additionally, we have clarified that all baselines in this section follow the same protocol, ensuring a fair and consistent evaluation framework.

---

### Official Review · Reviewer_7S2t · 2024-11-02

**Soundness:** 3
**Presentation:** 3
**Contribution:** 3
**Rating:** 6
**Confidence:** 4

**Summary:**

The paper presents DualTime, a novel multimodal language model designed for time series representation learning. It integrates both temporal and textual modalities to enhance the model's performance in disease detection and classification tasks. The authors argue that existing methods often overlook the unique information provided by text, which can lead to sub-optimal outcomes. By employing a dual adapter framework, DualTime aims to effectively explore the complementary information in multimodal inputs. The experimental results demonstrate that DualTime consistently outperforms other methods across various proportions of labeled data, showcasing its robustness and effectiveness.

**Strengths:**

1)This paper presents an innovative dual-adapter framework in the field of multimodal learning, treating text and time series modalities equally for the first time. This approach not only addresses the limitations of existing methods in handling multimodal data but also demonstrates the potential of effectively integrating both modalities' information for disease detection and classification. Notably, the introduction of learnable adapter tokens facilitates the extraction of high-level multimodal semantics, which is a novel contribution to the existing literature. The proposed problem has important research implications and application value.
2)The experimental design of the paper is rigorous, covering multiple real-world datasets (such as PTB-XL and TUSZ) and employing various evaluation metrics (accuracy, precision, recall, and F1 score) for comprehensive assessment. The results indicate that DualTime consistently outperforms other baseline methods across different proportions of labeled data, enhancing the credibility of the findings through systematic validation.
3)The paper is well-structured and logically coherent, with concepts articulated in an easily understandable manner. The authors clearly outline the research background, objectives, and significance in the introduction, and provide detailed descriptions of the design and implementation of DualTime in the methods section. Additionally, the use of figures and tables effectively supports the textual content, making complex concepts more digestible.

**Weaknesses:**

1) The paper mentions that the choice of adaptive token length and the number of network layers significantly impacts model performance, but lacks a systematic analysis of these parameter choices. It is recommended that the authors provide additional results on parameter sensitivity analysis to help readers understand how to optimize the model.
2) DualTime employs a dual-adapter design that treats the textual modality and temporal modality as primary and auxiliary modalities, respectively, facilitating dynamic fusion at a high level through learnable adapter tokens. This method shares a pre-trained language model, enhancing the capture of complementary information between the two modalities and improving the model's learning efficiency and performance. To further demonstrate the effectiveness and advancement of this fusion method, experiments should be conducted comparing it with other fusion approaches, such as simple concatenation, weighted fusion, and attention mechanisms.

**Questions:**

How does DualTime ensure that the integration of textual and temporal modalities does not lead to overfitting on specific datasets? What strategies are employed to enhance the model's generalization capabilities across diverse datasets, particularly in practical applications? Do the authors plan to extend this work to other types of time series data (beyond EEG and ECG), and if so, what adjustments would be necessary?

---

> ### Author Response · Authors · 2024-11-24
>
> We are most thankful for your assessment, and glad to communicate with you on your concerns:
>
> > **W1: Token length & Network layers Analysis**
>
> We have a sensitive analysis section in the Appendix A.6.  Here we quickly recap the result for you:
>
> The performance of our model tends to improve with an increase in the number of multimodal fusion layers. While the length of adaptation tokens has a relatively small impact.
>
>
> > **W2: Modality Fusion Mechanism**
>
> We have added experiments in the Appendix A.7 on different fusion strategies for the auxiliary and primary modalities within each adapter. The table below briefly presents the experimental results.
>
> | DualTime             | PTB-XL    | PTB-XL    | PTB-XL         | PTB-XL         | TUSZ      | TUSZ      | TUSZ           | TUSZ           |         |         |
> | -------------------- | --------- | --------- | -------------- | -------------- | --------- | --------- | -------------- | -------------- | ------- | ------- |
> |                      | Detection | Detection | Classification | Classification | Detection | Detection | Classification | Classification | Average | Average |
> |                      | Acc.      | F1        | Acc.           | F1             | Acc.      | F1        | Acc.           | F1             | Acc.    | F1      |
> | Adaptation Tokens    | 0.83      | 0.76      | 0.81           | 0.74           | 0.84      | 0.69      | 0.79           | 0.78           | 0.82    | 0.74    |
> | Simple Concatenation | 0.76      | 0.62      | 0.73           | 0.61           | 0.79      | 0.63      | 0.72           | 0.55           | 0.75    | 0.60    |
> | Attention Mechanism  | 0.77      | 0.63      | 0.75           | 0.66           | 0.79      | 0.65      | 0.75           | 0.59           | 0.77    | 0.63    |
> | Weighted Fusion      | 0.79      | 0.69      | 0.76           | 0.66           | 0.81      | 0.67      | 0.78           | 0.75           | 0.78    | 0.69    |
>
> We found that dynamic fusion through learnable adaptation tokens achieved the best performance, with an average accuracy of 82%. In contrast, simple concatenation had the poorest performance, with an average accuracy of 75%.  likely because it is a static method without learnable parameters, leading to weak generalization capabilities.
>
> The attention mechanism demonstrated the second-lowest performance, achieving an average accuracy of 77%. While it improves upon simple concatenation by introducing a self-attention mechanism, it treats modality tokens almost equally, failing to emphasize the primary and secondary modalities effectively. This lack of distinction causes the textual-primary module and temporal-primary module become similar, making it more challenging for the model to extract the unique information contributed by each modality.
>
> Weighted fusion performed second-best achieving 78% accuracy , perhaps because it can adaptively determine which modality is more important. However, weighted fusion may prioritize one modality over the other, potentially reducing the model's ability to fully extract valuable information from the less prioritized modality. This imbalance could limit the fusion's effectiveness in scenarios where both modalities contribute complementary and unique information. In contrast, the use of learnable adaptation tokens in two modules enforces a distinction between the primary and secondary modalities, guiding the model to focus more effectively on the primary modality.
>
>
> > **Q1: Overfitting and Model generalization**
>
> **Preventing Overfitting:** DualTime employs a shared frozen pre-trained language model across both adapters, ensuring that only lightweight, learnable adapter tokens are added to each layer. This significantly reduces the number of trainable parameters, minimizing the risk of overfitting, particularly on small datasets. As we analyze at efficiency evaluation section, DualTime has approximately 1.0 million trainable parameters—larger than PatchTST (0.7M) but significantly smaller than TimesNet (8.6M) on TUSZ classification scenario.
>
> **Generalization Across Datasets:** DualTime is designed without reliance on any dataset-specific domain knowledge, making it inherently adaptable to diverse scenarios. While the core idea of DualTime is to equally treat both modalities equally so that to fully leverage the rich complementary semantics of each modality, it is most suitable for scenarios where temporal and textual modalities are paired one-to-one with close relations (for example, the ECG signal of patient and the corresponding historical clinical report) and both modalities can provide rich decision-relevant information.
>
> **Future Work and Applicability to Other Domains:** We recognize the potential to extend this research beyond EEG and ECG to other forms of time series data. For this extension, adjustments may involve domain-specific preprocessing.

---

> ### Author Response · Authors · 2024-11-28
>
> To Reviewer 7S2t:
>
> Thank you very much for your reply! Please let us know if these responses meet your expectations. We are eager to engage in further discussions and continue improving our work.

---

### Official Review · Reviewer_2wTR · 2024-11-03

**Soundness:** 3
**Presentation:** 3
**Contribution:** 3
**Rating:** 5
**Confidence:** 2

**Summary:**

The paper introduces DualTime, a novel multimodal language model designed for time series representation learning. The key contributions and highlights of the paper are as follows:

Textual-Temporal Multimodal Learning Paradigm: The paper proposes a new paradigm that treats both time series and text modalities equally, aiming to fully leverage their complementary information. This paradigm is designed to capture the intricate interactions between different modalities. DualTime consists of two multimodal adapters: a temporal-primary multimodal adapter and a textual-primary multimodal adapter. Each adapter treats one modality as primary and enhances it with the other modality. This dual-adapter design allows each modality to serve as the primary modality, improving the model's ability to capture unique and shared semantics.

Comprehensive Evaluation: The paper includes extensive experiments, ablation studies, and sensitivity analyses to validate the model's effectiveness, robustness, and efficiency. It also explores the impact of different textual encoders and discusses the model's extensibility to more modalities.

**Strengths:**

Novel Paradigm: The introduction of a textual-temporal multimodal learning paradigm that treats both time series and text modalities equally is a significant departure from existing approaches, which typically prioritize time series as the primary modality. Specifically, the authors propose two novel designs:
(1) Dual Adapter Design: The dual-adapter architecture, where each adapter treats one modality as primary and enhances it with the other, is an innovative approach to multimodal fusion. This design allows for a more comprehensive understanding of the interactions between modalities.
(2) High-Level Multimodal Fusion: The use of learnable adaptation tokens to achieve high-level multimodal fusion within the top layers of the language model is a novel technique that enhances the model's ability to capture complex semantics.

Comprehensive Experiments: The paper includes extensive experiments on public real-world datasets, demonstrating DualTime's superior performance in supervised, unsupervised, and few-shot learning scenarios. The results show consistent improvements in accuracy and F1 score across various tasks.

**Weaknesses:**

Consider stronger baselines for text-based methods. GPT-4o and the LLaMA series models serve as robust baselines for text-based approaches. BERT and GPT-2 are now outdated in the realm of NLP. Given that GPT-2 has performed reasonably well, I am curious about the capabilities of GPT-4o. While these models do have more parameters, they are easily accessible and offer low latency. In practical applications, they are excellent candidates.

From my perspective, I recommend the author use state-of-the-art pre-trained checkpoints, such as LLaMA-3, for the experiments. GPT-2 is outdated and insufficient for validating the capabilities of modern language models. While the paper seems more suited to 2022, it is now 2024, and demonstrating success with GPT-2 does not effectively showcase its relevance to current models.

**Questions:**

N/A

---

> ### Author Response · Authors · 2024-11-24
>
> Thank you for your review, and we are glad to address all your concerns:
>
> > **W1: Stronger Text-based baselines**
>
>
> We have introduced LLaMA-3 and the medical language model ClinicalBERT as new baselines for comparison. Due to budget constraints, we were unable to include GPT-4o in our experiments. The table below summarizes the results of these additional evaluations. For further details, please refer to the updated manuscript.
>
> |              | PTB-XL    | PTB-XL    | PTB-XL         | PTB-XL         | TUSZ      | TUSZ      | TUSZ           | TUSZ           | Average | Average |
> | ------------ | --------- | --------- | -------------- | -------------- | --------- | --------- | -------------- | -------------- | ------- | ------- |
> |              | Detection | Detection | Classification | Classification | Detection | Detection | Classification | Classification |         |         |
> |              | Acc.      | F1        | Acc.           | F1             | Acc.      | F1        | Acc.           | F1             | Acc.    | F1      |
> | GPT2         | 0.72      | 0.58      | 0.73           | 0.62           | 0.72      | 0.50      | 0.64           | 0.58           | 0.70    | 0.57    |
> | BERT         | 0.70      | 0.53      | 0.73           | 0.62           | 0.72      | 0.49      | 0.59           | 0.40           | 0.69    | 0.51    |
> | Llama 3      | 0.73      | 0.60      | 0.74           | 0.55           | 0.72      | 0.63      | 0.66           | 0.47           | 0.71    | 0.56    |
> | ClinicalBERT | 0.73      | 0.53      | 0.74           | 0.55           | 0.72      | 0.66      | 0.67           | 0.43           | 0.72    | 0.54    |
> | DualTime     | 0.83      | 0.76      | 0.81           | 0.74           | 0.84      | 0.69      | 0.79           | 0.78           | 0.82    | 0.74    |
>
>
> It can be observed that LLaMA-3 outperforms GPT-2 and BERT, achieving an average accuracy that is 1% higher than GPT-2 and 2% higher than BERT. The performance gap among these models is relatively small, likely because the tasks in this study are simple, allowing medium-sized language models to effectively capture patterns in the data. Consequently, larger-scale language models do not exhibit a significant advantage.
>
> Notably, ClinicalBERT, a model specifically pre-trained with medical domain knowledge, achieves the highest performance among all text-based baselines. It surpasses LLaMA-3 by an average of 1% in accuracy. This superior performance can be attributed to the alignment between our medical datasets and ClinicalBERT’s domain-specific pre-training, which enables it to better understand and process the text data.

---

> ### Author Response · Authors · 2024-11-28
>
> To Reviewer 2wTR:
>
> Thank you very much for your reply! Please let us know if these responses meet your expectations. We are eager to engage in further discussions and continue improving our work.

---

### Official Review · Reviewer_LNUH · 2024-11-03

**Soundness:** 2
**Presentation:** 3
**Contribution:** 2
**Rating:** 5
**Confidence:** 3

**Summary:**

This paper introduces DualTime, a parameter-efficient dual-adapter language model designed for multimodal representation learning that integrates both time series and textual data. Unlike prior models that often prioritize time series or text as the primary modality, DualTime employs a balanced textual-temporal paradigm, enabling each modality to enhance the other. By using two adapters within a shared language model backbone, the model captures high-level cross-modal interactions, achieving improvements of 7% in accuracy and 15% in F1 score on the PTB-XL ECG and TUSZ EEG datasets in supervised learning settings.

**Strengths:**

This paper presents an new approach to multimodal time-series learning through the introduction of the DualTime model, offering a fresh perspective that could inspire further exploration in balanced multimodal learning. It is well-structured and clearly presents the problem setting and results.

**Weaknesses:**

### Benchmarking in Diverse Domains and Foundation Models
Although DualTime performs well on the PTB-XL and TUSZ datasets, these are both medical datasets. It would be interesting to know DualTime’s performance and generalizability on additional domains beyond healthcare. A broader selection of multimodal dataset from other fields would provide more robust evidence of DualTime's effectiveness in multimodal time-series learning. Additionally, evaluating the model's architectural adaptability with other language models or LLMs could further demonstrate its potential generalization across different foundation models.

**Questions:**

The current experiment results lack sufficient evidence to show DualTime's general "Time Series Multimodal" capability. Beyond the text and time-series modality, how does DualTime perform with other modality combinations, such as visual and time-series or in non-medical domains?  Experiment results behind these questions could be important to claim DualTime's Time Series Multimodal capabilities.

---

> ### Author Response · Authors · 2024-11-24
>
> Thank you for your insightful comments. Here are our point-by-point answers to your concerns.
>
> > **W1: Benchmarking in Diverse Domains**
>
> The core idea of our framework is to treat both modalities equally, fully leveraging the rich complementary semantics that each modality provides. DualTime is particularly suitable for scenarios where temporal and textual modalities are paired one-to-one with closely related information (e.g., a patient’s ECG signal and their corresponding historical clinical report), and both modalities contribute meaningful, decision-relevant information.
>
> To date, we have only been able to collect two publicly available datasets that ideally meet these requirements. And we conduct extensive experiments from multiple angles (i.e. supervised learning, unsupervised learning, few-shot learning) to fully utilize these dataset to prove the effectiveness of DualTime.
>
> In addition, we are open to extending our experiments to high-quality datasets from other domains, provided they are publicly accessible.
>
>
> > **Q1: Benchmarking in Diverse Modality**
>
> In this work, one of our primary motivations is to explore the potential of language models in multimodal learning involving time-series data. While our idea to treat each modality equally—can be extended to other modality pairs, such as time series and images, DualTime cannot be directly applied to a time-series–image pair. This limitation arises because its backbone is a language model, which excels on text modality modeling but not on image modality. In future work, we will continue to explore how to more effectively extract complementary information from time-series data and other non-text modalities.
>
>
> > **W2: Benchmarking in Diverse Foundation Models**
>
> Our choice of GPT-2 as the backbone is based on several considerations. First, the amount of data available in most time-series scenarios is relatively limited. To reduce computational costs and mitigate the risk of overfitting, we opted for a moderately sized language model instead of larger-scale models like LLaMA. Additionally, the baselines we referenced, such as UniTime, GPT4TS, and GPT4MTS, also utilize GPT-2 as their backbone. These studies validate GPT-2's capability in handling time-series data. We also want to compare with these baselines as fair as possible so we chose GPT-2 as the backbone for our model. In this way, we can better validate the effectiveness of our temporal-textual paradigm and adapter fusion mechanism instead of the choice of backbone.
>
> While we are unable to provide a backbone ablation study due to the time constraints of the rebuttal phase, we are conducting the experiments and will include the results as soon as possible or in the Appendix of our final revised manuscript.

---

> ### Author Response · Authors · 2024-11-28
>
> To Reviewer LNUH:
>
> Thank you very much for your reply! Please let us know if these responses meet your expectations. We are eager to engage in further discussions and continue improving our work.

---

### Official Review · Reviewer_P6LV · 2024-11-09

**Soundness:** 3
**Presentation:** 3
**Contribution:** 2
**Rating:** 5
**Confidence:** 3

**Summary:**

The paper introduces DualTime, which employs a dual-adapter architecture such that time-series and text modalities can each serve as primary modes. It introduces learnable tokens into the top layers of the LM backbone to facilitate multimodal fusion. The method leads to improvements over baseline methods, showing the method's effectiveness in leveraging information from both modalities.

**Strengths:**

1. Detailed experiments are conducted to present the effectiveness of the method, including both supervised fine-tuning and unsupervised learning with linear probing. The experiments show the method is able to learn general representations for the given task.
2. Few-shot transfer is further investigated, proving the generalizable features learned from the method, though the experiments are conducted in an in-dataset setting and it is unclear if it is useful for cross-domain transfer.

**Weaknesses:**

The novelty of the adapter and fusion mechanism is more or less limited. From the results, it seems like the gap between DualTime (text), where text serves as a primary mode, and DualTime is small. What are the implications of this pattern?

**Questions:**

What is the design choice for the LMs? e.g., why does the main backbone (GPT-2) need to be different from the text encoders used to encode secondary text inputs (BERT)? Are there any ablations about the main backbone?

---

> ### Author Response · Authors · 2024-11-24
>
> We are most thankful for your assessment. We have revised the paper based on your comments and highlighted the changes in red for clarity.
>
> > (1) **W1: Novelty of DualTime**
>
> DualTime introduces several key innovations. Beyond employing a fusion mechanism based on learnable adaptation tokens, DualTime proposes a synchronized learning strategy to jointly model the time-series and text modalities. Unlike prior works that primarily focus on the time-series modality, DualTime emphasizes the auxiliary modality, which provides rich and complementary information. However, assigning equal importance to different modalities poses challenges related to model complexity and computational cost. To address these issues, DualTime introduces a lightweight shared-backbone framework that enables efficient modality fusion using only a small number of learnable adaptation tokens.
>
> > (2) **W2: Performance Gap & Pattern Implication**
>
> DualTime (text) outperforms DualTime (time), demonstrating that the language model effectively extracts more task-relevant information when the text modality serves as the primary modality. This is not surprising, as both are sequential data, yet language models are inherently better at handling the text modality compared to time series. However, DualTime (text) still falls short of DualTime by an average accuracy gap of 3% in supervised learning and 4% in unsupervised learning, underscoring the importance of balanced learning across both modalities.
>
> > (3) **Q1&Q3: Choice of LLM &  Backbone Ablation**
>
> Our choice of GPT-2 as the backbone is based on several considerations. First, the amount of data available in most time-series scenarios is relatively limited. To reduce computational costs and mitigate the risk of overfitting, we opted for a moderately sized language model instead of larger-scale models like LLaMA. Additionally, the baselines we referenced, such as UniTime, GPT4TS, and GPT4MTS, also utilize GPT-2 as their backbone. These studies validate GPT-2's capability in handling time-series data. We also want to compare with these baselines as fair as possible so we chose GPT-2 as the backbone for our model. In this way, we can better validate the effectiveness of our temporal-textual paradigm and adapter fusion mechanism instead of the choice of backbone.
>
> While we are unable to provide a backbone ablation study due to the time constraints of the rebuttal phase, we are conducting the experiments and will include the results as soon as possible or in the Appendix of our final revised manuscript.
>
> > (4) **Q2: Relation between Backbone and Text Encoder**
>
> Regarding the choice of the text encoder, it does not necessarily have to differ from the main backbone. In Section 3.5, Explorations on Model Design, we analyzed the impact of various text encoders, including GPT-2, BERT, and ClinicalBERT. Our findings reveal that BERT-based textual encoders consistently outperform GPT-2-based encoders.

---

> ### Author Response · Authors · 2024-11-28
>
> To Reviewer P6LV:
>
> Thank you very much for your reply! Please let us know if these responses meet your expectations. We are eager to engage in further discussions and continue improving our work.

---

### Author Response · Authors · 2024-11-24

Dear Reviewers, ACs, SACs, and PCs,

Thanks for your insightful feedback on our manuscript: DualTime: A Dual-Adapter Language Model for Time Series Multimodal Representation Learning!

Within this short rebuttal period, we did our best to address all your concerns, and we have incorporated our responses into the updated version.

If you have any questions towards our responses, please feel free to let us know.

Sincerely,

Authors

---

### Meta-Review · Area_Chair_Vzvg · 2024-12-18

**Metareview:**

The paper introduces DualTime, a multimodal language model designed for time-series and textual representation learning. Its key innovation lies in a dual-adapter architecture, where each modality (time series and text) serves as the primary mode in separate adapters, enabling balanced cross-modal interaction and effective fusion. This architecture allows the model to fully leverage the complementary information of the two modalities, improving predictive performance. Claimed contributions of the work included: 1) a new paradigm of textual-temporal multimodal unsupervised learning with both time-series and text modalities equally prioritizes, 2) parameter-efficient design with adapter, 3) ablations and empirical results showed the efficacy of proposed method.


Strength of this paper

- Ablations and empirical results showed the efficacy of proposed method.
- The approach is computationally efficient
- Code is shared, ensuring the reproducibility of experiments.
- The proposed approach provides a fresh perspective on balanced multimodal learning, inspiring further exploration in the field; the work also advances the research in multimodal representation learning.


Weakness of this paper

Several reviewers raised few concerns/limitations of this paper. By addressing these limitations, the paper could strengthen its experiment and expand impact.

- Some of the chosen baselines (e.g., BERT and GPT-2) are outdated. Dataset used to evaluate DualTime (only on two medical datasets, PTB-XL and TUSZ) is also limited. These limit the evidence for the generalizability and efficacy of proposed approach. Broader evaluations on multimodal datasets from diverse domains and compare with modern models, such as GPT-4o or the LLaMA series, would strengthen the paper’s claims about its effectiveness in multimodal time-series learning. Several choices in experiments such as freezing the language model backbone in baseline methods or selection of datasets are not well-justified and may misrepresent the experimental findings. Parameter and ablation analysis are limited (e.g., lack of analysis on adaptive token length and the number of network layers impact on performance, not explore scenarios where unsupervised learning is excluded or the delta between configurations with and without unsupervised learning)
- Novelty is also a concern. Adapter + fusion mechanism has been widely studied in the community. The paper does not sufficiently compare the proposed approach to other established methods, such as simple concatenation, weighted fusion, or attention mechanisms, and convincingly articulate the position of proposed approach and added value.

**Additional Comments On Reviewer Discussion:**

In addition to above weaknesses, reviewers also raised some other weaknesses (e.g., clarity in manuscript, notations and formulas) and improvements during rebuttal. Although some of the weakness have been improved / somewhat addressed during rebuttal session (e.g., further explanation or clean up the manuscript, more experiment results added), overall review rating was not raised significantly, and the rating is still at borderline.  I think the session is too short and some weaknesses are hard to address in such a short period of time (e.g., comprehensive experiments, better position of the paper and work compared to established works). I would like to see a more comprehensive modification to systematically working on these suggestions. Given the high bar of ICLR, I think the paper is still of limited interests to the audience , and thus I recommend to reject the paper, and the authors to re-work on these weakness and re-submitting to future conferences.

---

### Decision · Program_Chairs · 2025-01-22

Reject